# Robust minimally-invasive microfabricated stainless steel neural interfaces for high resolution recording

Zabir Ahmed [1], Ibrahim Kimukin[1], Vishal Jain[1], Kate Gurnsey[2], Tobias Teichert[2] & Maysamreza Chamanzar [1,3,4] ✉

Understanding brain function and developing effective neurotherapeutics require high-resolution electrophysiologyical recording across large primate brains. To achieve this goal, minimally invasive, compact, long, and high-density implantable neural probes are needed. Conventional silicon-based probes, designed for rodents, cannot be directly scaled to larger brains because silicon's brittleness restricts long, high–aspect-ratio designs. Stainless steel is a biocompatible, resilient, and less brittle alternative for making neural probes, though its microfabrication is less explored. Here, we introduce "steeltrodes", customizable microfabricated stainless steel neural probes enabling high-density multi-layer electrode integration. We demonstrate 8 cm long, ~300 μm wide probes, featuring rigid shanks with optional flexible cables for in vivo high-resolution laminar recording from macaque auditory cortex. In rats, these probes can be safely implanted through intact dura with minimal cortical damage. High-fidelity recordings of single units and local field potentials in macaques highlight the potential of steeltrodes for translation to human applications, enabling both inter- and intraoperative neural recordings.

Over the last decades, novel manufacturing and microfabrication processes have revolutionized neural probe technology for rodents[1–5]. Based in large parts on the adaptation of silicon as the material platform of choice, it has been possible to massively increase recording channel density using high-resolution lithography and microfabrication processes and to add new functionalities such as optical stimulation and imaging and chemical sensing[5–8]. However, silicon has a low fracture toughness and therefore, it is very brittle. As a result, it has been challenging to adapt these new tools to the larger brains of primates. Probes for primates need to be an order of magnitude longer (4–15cmlong as opposed to just a few mm-long rodent probes), while at the same time, maintaining a very small cross-section to minimize tissue damage[9]. As a result of its high-aspect ratio, a typical NHP (non-human primate) probe requires substantial rigidity and robustness to

endure the stresses of implantation and post-implantation when implantedinserted without rigid inserters.

Nevertheless, silicon-based neural probes have been used in NHPs and humans. However, the implantable length of those probes is usually shorter than 3 cm which limits the recording to the superficial cortical regions of large animals[7,10,11] and humans[12]. There are also reports of silicon probes shattering during insertions into human brains[12]. Michon et al. reported thin silicon probes with 4-cm-long shanks integrated with a custom-designed microdrive array for recording from large ensemble of neurons in rats[13]. These thin silicon probes are somewhat flexible. However, the implantable shank length was limited to 7mm due to physical constraints of the custom microdrive array. Recently, a silicon-based high-density NHP probe with 45 mm shank has been reported for recording from deep brain

[1]Department of Electrical and Computer Engineering, Carnegie Mellon University, Pittsburgh, PA, USA. [2]Department of Psychiatry and Bioengineering, University of Pittsburgh, Pittsburgh, PA, USA. [3]Department of Biomedical Engineering, Carnegie Mellon University, Pittsburgh, PA, USA. [4]Neuroscience Institute, Carnegie Mellon University, Pittsburgh, PA, USA. ✉e-mail: mchamanzar@cmu.edu

structures in macaques[14]. However, due to silicon's inherent fragility, the use of specialized equipment and implantation hardware are essential for ensuring proper alignment of such probes to reduce the likelihood of the probe breaking during insertion into the deeper areas of the brain[9,15].

In addition to silicon, polymer materials have also been used to implement neural probes for rodents and large animals. Compared to silicon, polymers are much less vulnerable to fracturing and much more flexible. Polymer-based neural probes have been shown to minimize tissue damage upon implantation due to their significantly smaller Young's modulus[16–19]. However, lower rigidity of polymeric probes makes them more difficult to implant, which necessitated designing new schemes for implantation, including temporarily stiffening neural probes by coating bioresorbable materials to overcome the implantation force without buckling[20,21]. Pothof et al. reported microfabricated polymer-based Stereo-electroencephalography (SEEG) probes with high-density electrodes on a long implantable shank of 3 cm[22]. In this work, first, planar flexible polymer neural probes are microfabricated and then manually assembled to form a cylindrical shank, infilled with an epoxy. These neural probes have many electrodes (32–128) and the cylindrical cross section is rather large (~0.8 mm diameter). This work exhibits an interesting combination of microfabrication process and manual assembly towards realizing high-density neural probes for large animals.

In summary, the material platform developed for rodent probes cannot be directly translated to fulfill the challenging design requirements of high density, compact, yet robust neural probes for primates and humans. This gap calls for new material platforms and novel microfabrication processes. Given its highly desirable mechanical and biomedical properties, austenitic stainless steel is an excellent candidate for implementing large aspect ratio neural implants. Stainless steel 316 has a fracture toughness of 112–278 MPa$\sqrt{m}$[23], nearly two orders of magnitude higher than that of silicon, which is only 0.7–0.9 MPa$\sqrt{m}$[24]. Fracture toughness, $K_{Ic}$ is a material property that characterizes the material's robustness to fracturing. Fracture toughness is a function of stress ($\sigma_c$) required to fracture the bulk material for a given crack length ($a$), as described in this equation: $K_{IC} = \sigma_c\sqrt{\pi a}$. Therefore, for the same crack length ($a$), compared to silicon, which has a much lower fracture toughness[24], it would take almost two orders of magnitude higher stress to fracture a stainless steel device. As a result, a neural probe with extreme aspect ratio fabricated on stainless steel will be significantly less likely to fracture compared to a silicon neural probe. Therefore, stainless steel neural probes would be more robust and resistant to fracturing. In addition, stainless steel 316 has a high Young's modulus of 190–203 GPa[23], which is comparable to that of silicon (140–180 GPa[24]). Since the overall stiffness is determined by the device geometry and the material elastic modulus, a stainless steel probe offers a comparable level of stiffness to a silicon probe with similar dimensions, but with much less vulnerability to fracture. Therefore, unlike polymer-based probes, stainless steel probes can be implanted reliably without the need for additional stiffener layers.

Stainless steel also has high corrosion resistance and is biocompatible. It has been used to manufacture biomedical implants such as prosthetics and coronary stents[25–29]. Most devices fabricated using this material platform are made using macro-scale manufacturing and machining techniques. Commercially available stainless steel neural probes for NHP are manually assembled (Plexon Inc.). Such neural probes are used for NHP experiments because of their robustness and reliability. The manual manufacturing of such neural probes puts constraints on the density of microelectrode channels (e.g., currently, up to 64 channels with a minimum pitch of 50 μm) and limits the yield. In addition, the probes are very expensive[30] and while these probes are typically of very high quality, they are nevertheless subject to human error due to the non-scalable hand-assembly processes.

Optimized scalable microfabrication and micromachining processes are necessary to leverage the excellent material and mechanical properties of stainless steel to design miniaturized biomedical devices such as high-channel density neural probes with micron-scale features on stainless steel. In the case of silicon probes, fabrication and micromachining process had benefitted from decades of research and development in the MEMS/NEMS (Micro-/Nano-Electromechanical Systems) and CMOS (Complementary metal–oxide–semiconductor) electronic industries[1,31–33]. However, the same processes cannot be readily translated to stainless steel. Moreover, the micro- and nano-fabrication processing for stainless steel is quite challenging and comparatively underdeveloped and underexplored. In this paper, we show that it is possible to overcome these challenges and enable micro- and nanofabrication for stainless steel. Robustness of stainless steel material combined with the unmatched scalability of planar microfabrication paves the way for realization of different biomedical devices with extreme aspect ratios, not currently feasible using traditionally used material platforms. These devices will be robust and reusable, customizable and can be mass produced.

In this paper, we showcase the material platform and a scalable microfabrication process by realizing high-density, ultra-compact stainless steel neural probes (aka *steeltrodes*), that are long enough to record from anywhere in the NHP brain. Recording neural activity across different areas of brain with high resolution and minimal damage to the brain tissue, especially in larger animal models and humans has remained elusive mainly due to the lack of a robust and minimally invasive implantable neural interface platform that can be mass produced and reliably used. The steeltrodes discussed in this paper provide a path towards high-resolution distributed neural recording from various brain areas, as shown in Fig. 1a, where multiple steeltrodes are implanted at different depths within the brain. Each of the penetrating neural probes is a hybrid polymer-stainless steel neural probe with a rigid shank that can be designed to be fully stiff (Fig. 1b) or monolithically connected to a flexible cable (Fig. 1c, d) to route the recorded signals to the backend circuitry, while minimizing the tethering force and damage to the brain tissue due to brain micromotions[34]. Steeltrode features high-density microelectrodes with highly customizable size and arrangement (Fig. 1e–g). This high degree of flexibility in the design, combined with a repeatable high-throughput microfabrication process, enables a plethora of applications, ranging from basic science research on rodent and NHP models to acute and chronic neural interfacing with human neural tissue for clinical and therapeutic applications. As a demonstration of the capabilities of the platform, steeltrodes with long implantation length (8 cm) with a very thin cross section of 140 μm × 280 μm with high-density microelectrodes (16–100 channels, with pitches ranging from 25 μm to 2.5 mm) were fabricated and tested in macaque auditory cortex which is located deep in the lateral fissure and is notoriously difficult to reach. With an implanted steeltrode, we were able to reliably record single-unit and multi-unit activity as well as local field potentials (Fig. 1i). Given their compactness, high-density design of electrodes and their exquisite robustness, these steeltrodes can be translated to clinical applications such as intraoperative neural recording from human subjects, in the future.

While our steeltrode design was optimized for fabrication of NHP probes, it can be readily scaled down to make mm-long probes for rodents (inset of Fig. 1b), where we can still benefit from the robustness of stainless steel over conventional silicon-based neural probes to minimize the risk of failures during implantation and experiment and introduce a robust and re-usable neural recording platform. Despite having smaller aspect ratios, rodent silicon neural probes are still susceptible to fracturing owing to the material properties which makes the manipulation of such probes challenging. Moreover, implantation of these conventional silicon neural probes usually requires removing

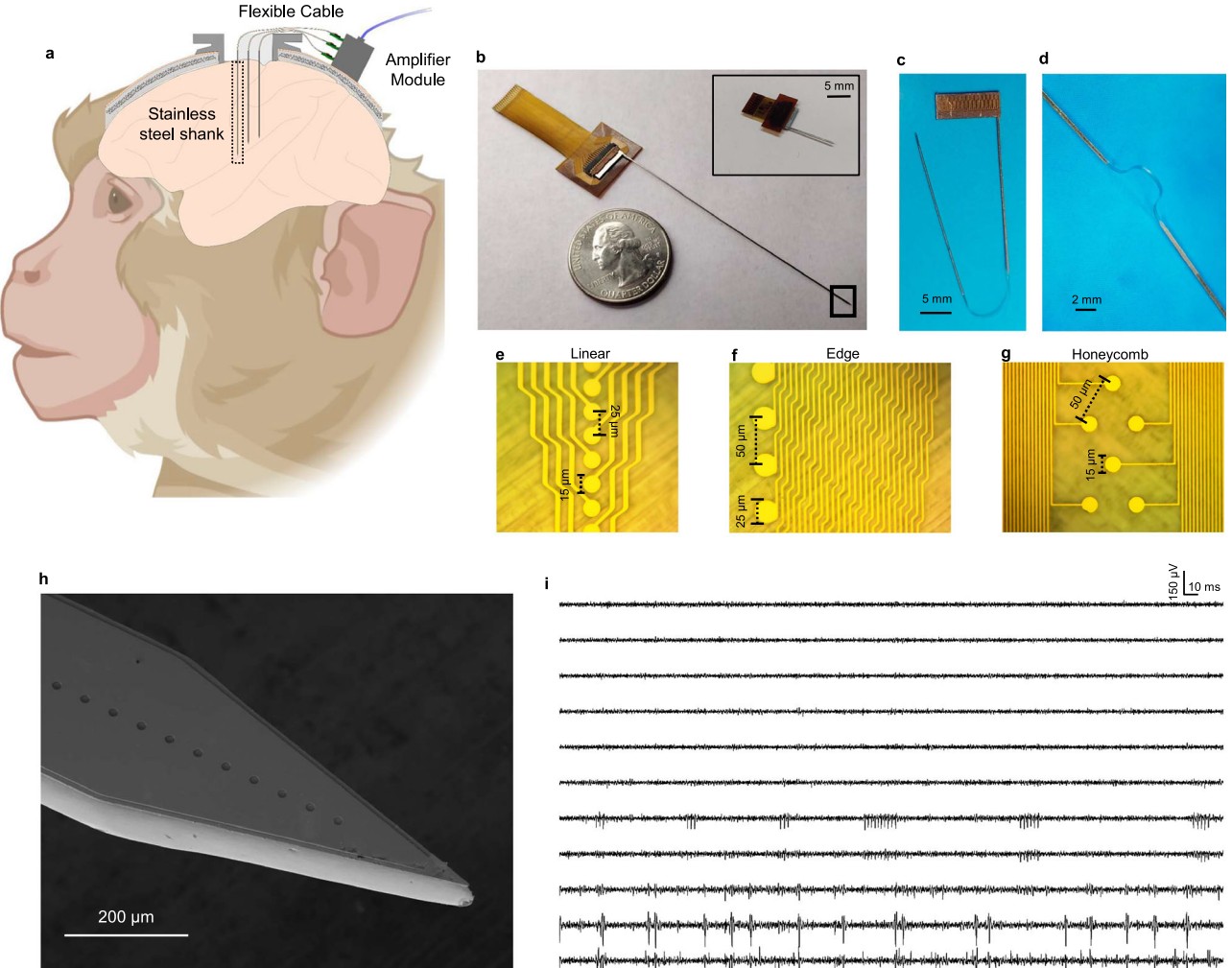

**Fig. 1 | Overview of steeltrode, a highly customizable microfabricated stainless steel neural probe. a** Schematic of implantation of multiple hybrid stainless-steel polymer neural probes in non-human primate brain (created in BioRender. Chamanzar, M. (2025) https://BioRender.com/qms5tta). **b** Photograph of a fully rigid steeltrode with 8-cm-long shank designed to record from deep brain structure in NHP. Inset shows a shorter variant of steeltrode suitable for measurements in smaller animals such as rodents (**c**) Photograph of a steeltrode with a flexible tether cable **d** Magnified image of the flexible tether cable. **e**–**g** Micrographs of different arrangements of recording microelectrodes on steeltrode. **h** SEM image of the tip of steeltrode with a cross section of 140 μm × 280 μm consisting of 32 recording microelectrodes, each with 15 μm diameter, and spaced 50 μm apart. **i** High-pass filtered multichannel neural signal captured from macaque.

the protective dura layer, that can cause trauma and exposing a region of the brain larger than the probe cross section, that can compromise the cranial pressure. In contrast, since steeltrodes are resilient against fracture, they can be reliably inserted into the neural tissue multiple times and their handling and manipulation is safe and repeatable. Moreover, these robust neural probes can be implanted through rodent dura, obviating the need for removing dura, thus reducing damage to the superficial layers of the neural tissue. Therefore, in addition to enabling high-fidelity recording from deep brain regions of large animals, the microfabrication process on stainless steel discussed in this work can also enable manufacturing of mechanically robust stainless steel probes with smaller implantable shanks for recording from small animals without the need for surgical removal of dura.

## Customizable design parameters of Steeltrode architecture

Since the steeltrodes are microfabricated using advanced lithography and micromachining techniques, their design can be easily customized. The size and geometry of the neural probe as well as the arrangement, shape and size of the electrodes can be designed for the intended application. Here we discuss design considerations to choose these parameters.

## Different lengths of implantable shank

Implantable length of the shank is an important design parameter for neural probes. The length requirements are often dictated by the depth of the target region within the brain, the approach angle of the probe during implantation and the dimensions and locations of the implantation chamber, as well as the hardware used for probe fixtures and the microdrive system for advancing the probe into the neural tissue. Neural probes with only a few centimetres of shank length can be sufficient to record from superficial layers of brain in non-human primates. For rodents, this length can be even much smaller, on the order of just a few millimeters. However, to record from deeper regions of the brain, neural probes with much longer implantable shanks are necessary both for NHPs and rodents. For example, auditory cortex in *macaca mulatta* is located 1–2.5 cm below frontal cortex when approached perpendicular to the cortical sheet as is common for laminar recordings. Moreover, for macaque electrophysiology, chronic recording chambers and implantation grids are used for probe implantation. In addition to the depth of the target below the surface of brain, the additional length taken up by the fixtures to hold and insert the neural probe should also be considered to determine the appropriate length of the neural probe. Based on these design

constraints, we have designed and implemented stainless steel probes with >8-cm-long implantable shanks that can probe auditory cortex, or any other neural structure in the macaque brain. The cross section of these probes is designed to be 140 μm × 280 μm to keep the number of recording channels high (i.e., up to 100), while minimizing the tissue damage. With these dimensions, the aspect ratio of our longest steeltrode for NHPs (defined as the ratio of shank length to the smallest cross-sectional dimension) would be ~570. This is a very high-aspect ratio compared to existing probes for NHPs, for example, Plexon V probe and Neuropixel (NHP-Long) probe have aspect ratios of ~307[30] and ~360–500[35], respectively. For an 8-cm-long, 140-μm-thick, and 260-μm-wide steeltrode, the buckling force will be 40 mN, whereas the buckling force for a silicon probe with the same dimensions would be 32 mN. While the buckling force for probes manufactured in silicon and stainless steel may be similar, microfabricated silicon probes with shanks of such high-aspect ratio will be highly susceptible to fracturing. It has been reported that fracture strength of microfabricated silicon devices can vary widely depending on the processing conditions, size, and geometry[36]. Specifically, as the surface area of the device is increased, the fracture strength is reduced. For a long-aspect ratio microfabricated silicon neural probe with 8-cm-long shank and 140 μm × 280 μm cross section, the fracture strength on the stressed surface can be estimated to be ~300–500 MPa[36]. Finite Element Method (FEM) analysis of clamped-free silicon cantilever with similar dimensions reveals that ~5 mN applied at the free end induces stress surpassing the fracture strength, thus causing it to break. Whereas for a steeltrode with similar dimensions, such a force will only lead to elastic deformation of the structure, meaning that not only the probe will not break, but it will also regain its original shape after the force is released. This is because stainless steel (SS316, Hardness C44) used for implementing our steeltrodes has a high yield strength of 1.23 GPa (for 2% elongation). Therefore, a much higher force of ~40 mN would be required to induce plastic deformation of steeltrodes, according to our simulations. A significantly larger force would be required for stainless steel probe to fracture. Supplementary Fig. S1 shows the estimated forces required for fracturing silicon probe and yielding stainless steel probes with different lengths with a cross section of 140 μm × 280 μm. Therefore, stainless steel probes with extreme aspect ratios can sustain very high forces without fracturing compared to a silicon probe of the same dimension that can easily break with a much smaller force. In the example mentioned above, the steeltrode designed for recording from NHP brain can sustain a force (40 mN) before it deforms, which is about an order of magnitude larger than the force (5 mN) that a silicon probe of the same dimensions can tolerate before it breaks. We should also note the stark difference that beyond these force thresholds, the silicon probe shatters into many small and sharp pieces causing serious complications if this happens within the brain tissue, whereas the steeltrode only deforms and bends, thus causing much less and often reversible damage to the neural tissue. While implantation forces required to penetrate brain tissue is limited to few milli-Newtons, a neural probe can potentially be subjected to much higher forces during handling, implantation and the actual experiment. Chung et al. have reported multiple instances of fracturing of silicon neural probes during and after insertion in human brain. Forces acting on the probe due to large lateral movements were attributed to some of these breakages[12]. In addition, superficial regions of the brain over the targeted implantation region in primates can get calcified over time. In such cases, attempted implantation can induce much higher axial force leading to buckling and breaking of the silicon neural probes. The high fracture toughness of stainless steel makes the steeltrodes highly robust against accidental damage.

For recording from rodent brain, especially from cortical layers, 1–2 mm shank length is usually sufficient and existing silicon probes can serve the purpose. However, to probe deeper regions such as subthalamic nucleus (STN) in rats which is located about 8 mm below the surface of the brain, longer probes (>1 cm) would be needed. We can adapt our microfabrication process on stainless steel to implement mechanically robust steeltrodes with cm-long implantable shanks, which are suitable for recording from rodents and other small animals. Given the high Young's modulus and high fracture toughness of stainless steel, steeltrodes can sustain high forces during implantation without buckling or fracturing. As a result, steeltrode itself can be used to puncture dura of small animals during implantation, without the need for surgical removal of dura. The ability to puncture through the intact dura layer without surgical removal is critical both for retaining the cranial pressure and for reducing the volume of removed tissue for probe implantation.

Use of microfabrication processes allows high degree of customizability of the device length. Lengths and shape of the shanks can be defined with micrometer precision. The maximum length of the device is essentially limited by the diameter/size of the stainless steel substrate/wafer used in the microfabrication process. Most standard microfabrication facilities can accommodate wafer sizes up to 6-inch diameter in the tools used for micropatterning, deposition and etching of thin films. Probes with up to 18-cm-long shank can be fabricated on an 8-inch wafer. Using a 4-inch wafer process, we have demonstrated 8-cm-long steeltrodes, which can reach deep brain regions in macaques.

## Different sizes of recording electrodes

Size and arrangement of the recording microelectrodes in a neural probe is highly specific to the intended application and neurobiological experiment. In general, smaller microelectrodes are preferred for cellular-scale electrophysiology. Electrodes with large surface area are often preferred for recording local field potentials (LFPs). Viswam et al. have reported significant decay in the recorded signal amplitude for electrodes larger than $20 \times 20$ μm² due to spatial averaging when the signal source is in its close vicinity (i.e., within 20 μm range)[37]. Therefore, electrodes with sizes smaller than $20 \times 20$ μm² area are particularly suitable for recording extracellular action potentials from nearby neurons. On the other hand, local field potentials have higher signal amplitudes and larger spatial spread and can be easily recorded using larger electrodes. Stereo-EEG probes with large surface area electrodes have been used for epileptic source localization in human subjects from the recorded field potentials. Moreover, larger electrodes have lower electrochemical impedances leading to smaller thermal noise and reduced signal attenuation due to input impedance of the recording amplifier. Despite the higher spatial resolution, the smaller microelectrodes are at a relative disadvantage due to their higher electrochemical impedance. To reduce the associated thermal noise and signal attenuation associated with this large impedance, surface modifications such as depositing Pt-black or PEDOT: PSS have been used to achieve significantly reduced electrochemical impedance to enable high signal-to-noise-ratio (SNR) recording with smaller microelectrodes.

By leveraging high-resolution photolithography to pattern features for metal traces on stainless steel, we can manufacture steeltrodes with a wide range of electrode sizes. We have fabricated probes with electrodes as small as 10-μm diameter and as large as 40-μm diameter. The devices with small electrodes were used for probing spontaneous single and multi-unit activity from macaque auditory cortex, while the larger electrodes can be utilized to record LFPs in macaque auditory cortex.

The arrangement of the microelectrodes on the probe is also influenced by the objective of the neurobiological experiment. Tetrode arrangement of electrodes can potentially enable localization of neurons from the recorded single unit activities across the set of 4 electrodes. Linear electrode arrays are the most commonly available configuration for commercially available neural probes. These linear arrays can either be positioned along the center of the shank or along

the edge of the shank. Particularly for probes with relatively wide shanks, Fiáth et al. have reported significantly higher single unit yield and signal amplitude from electrodes along the edge compared to the electrodes arranged on the center of the shank. These results are reported for high-density silicon probes implanted in rat neocortex[38]. For steeltrodes, the presented microfabrication process allows us to realize neural probes with customizable electrode arrangements. We have implemented linear arrays with central and edge electrodes, as well as a honeycomb arrangement of electrodes, both for rodent and NHP steeltrodes. We have also demonstrated steeltrodes with different electrode diameters on the same shank.

Similar to electrode dimensions and arrangements, the density of the microelectrodes can also be highly customized. An array of microelectrodes with small inter-electrode separation can maximize the likelihood of recording single unit activity of neurons in the vicinity of the probe. Using a simplified point-current source, Vishwam et al. have reported that an electrode pitch of larger than 32 μm can lower the chances of picking up signals from the source with high SNR, whereas a smaller than 20 μm pitch can greatly increase the probability of recording the signals from the sources with minimal attenuation[37]. We have demonstrated a steeltrode design with an inter-electrode pitch of as small as 25 μm for 10-μm diameter electrodes for high spatial resolution single unit recording. On the other end of the spectrum, we have also manufactured probes with sparsely distributed electrodes with larger diameters (30–40 μm), with pitches ranging from 1.5 to 2.5 mm. These particular devices can be used to record LFPs from different layers of the brain. The active area of the shank consisting of the sparsely arranged electrodes can span up to 4 cm. The use of a planar microfabrication process on stainless steel allows parallel manufacturing of a large number of probes with different designs on the same stainless steel wafer.

## Rigid versus flexible probes

Brain tissue is soft with an elasticity of ~1 kPa[5,39], orders of magnitude smaller than the Young's modulus of rigid materials such as silicon and stainless steel. When rigid probes made of silicon or stainless steel are implanted into the brain tissue and anchored to skull, brain tissue can be damaged. Relative micromotions between the brain tissue and the skull is another factor that can exacerbate the tissue damage and potentially affect the long-term recording viability for skull-fixed rigid probes. The magnitude of such brain micromotions due to respiration and vascular pulsation can be in the range of 4–30 μm in rodents[40]. In monkeys and humans, the range of motion can be even higher[41]. In non-human primates, Chauvière et al., have demonstrated superior spiking activities and LFP recording with floating probes compared to skull-fixed probes during the first weeks of recording[42]. However, the authors concluded that in long-term, devices with both types of fixtures could record stable neuronal signals. For floating probes with flexible tether cables, one of the main drawbacks is the difficulty in adjustment of the depth of insertion post-implantation. Therefore, in addition to acute recording, rigid stainless steel probes can potentially be used for stable long-term recording.

Free-floating implants with rigid shanks have been shown to illicit less foreign body response when compared to a tethered probe in rodents over a time scale of 1–4 weeks[34]. Therefore, to minimize the tissue damage in chronic recording, neural probes with rigid shanks that are mechanically decoupled from skull fixtures via flexible cables can also be advantageous for large animals as well. Apart from tissue damage, brain motion relative to skull-fixed rigid neural probes can lead to movement artifacts which can negatively affect the single-unit yield. Chung et al. have reported ~250 μm of relative movement between the brain tissue and skull-fixed silicon neural probes in human brains[12]. Therefore, mechanically decoupled free-floating rigid probes with a flexible tether cable can potentially compensate for the position drift and improve single unit yield.

In the proposed hybrid stainless steel neural probe architecture presented in this paper, steeltrodes can be manufactured with completely rigid shanks or with monolithic flexible tether cables connecting the rigid backend to the rigid implantable shank. Since the flexible cable can be released from stainless steel using an electrochemical etching process, our completely rigid probes can be easily transformed into a hybrid rigid-flexible neural probe, by selectively etching a portion of the stainless steel shank post-fabrication. This way, the flexible cable will be monolithic with the architecture of the steeltrode. This is a great advantage over the conventional methods of manufacturing the rigid probes and flexible cables separately and packaging them together, resulting in a more streamlined and scalable fabrication process.

## Scalability

To record from a large span of brain areas with high spatial resolution, we need a large number of microelectrodes that are densely distributed. However, increasing the number of electrodes poses some challenges in terms of the physical dimensions of the neural probes. For passive electrical neural probes, scaling up the number of channels means increasing the number of traces connected to each recording electrode. Accommodating these large number of traces while keeping the probe cross section small requires ultra-high resolution lithography processes, which can be costly and has limited throughput of production, reducing the fabrication yield. In addition, with increasing density of metallic traces (Pt/Au/Pt stack or Cr/Au/Pt stack), in particular for probes with long shanks, careful design optimization is required to ensure minimal crosstalk between adjacent traces. High-resolution lithography techniques, i.e., deep-UV lithography and electron-beam lithography, have been used to pattern metallic traces with submicron inter-trace gaps and widths. Metallic traces with a width of ~250 nm have been demonstrated for rodent probes[43]. However, reduced interconnect gap can lead to increased crosstalk between adjacent traces. Reducing the trace width can offset the increased crosstalk from reduced interconnect gap. While neural probes with such ultra-narrow metal interconnects have been successfully demonstrated in rodents, for neural probes intended for use in non-human primate brain with more than 4-cm-long shanks, such an ultra-small width of the metallic wires may lead to increased trace resistance, which in turn can lead to increased signal attenuation from the recording sites and increased thermal noise. The thickness of the wires can be increased to compensate for this effect; however, increasing the wire trace thickness would in turn lead to increased crosstalk[44]. Apart from these design considerations and trade-offs, ultra-high-resolution lithography techniques such as electron-beam lithography are serial, rendering them cost-prohibitive and not suitable for manufacturing scalability compared to conventional photolithographic processes that can be highly parallelized. Using a conventional photolithographic process (with a lamp at the wavelength of 320-nm), we have demonstrated successful patterning of 1-μm metal features on polymer layers deposited on stainless steel substrate. Such high-density metallic features can be used to realize more than 100 channels on a 250-μm wide stainless steel probe.

## Vertically integrated multilayer traces

Vertically integrated multilayer traces are another approach for drastically scaling up the channel count for monolithically fabricated neural probes. Instead of drastically miniaturizing the interconnect widths and gaps, traces can be distributed across multiple layers interleaved by thin (only a few μm) insulation layers. Using this approach, interconnect traces with relatively larger feature sizes can be implemented by using cost-effective and scalable photolithographic processes, and at the same time, channel count and density can be significantly scaled up, while retaining a small probe cross section. Rodent neural probes with

up to 4-layer metal trace have been demonstrated in both Poly-imide and Parylene-based architectures[19,45–47]. Similarly, double-sided SU-8-based probes using a 2-layer metal process have also been reported[48]. The thickness of insulation layers between sub-sequent metal layers is an important design consideration for multilayer metal processing in neural probes. Liu et al. have reported less than 1.5% crosstalk at 1 kHz between vertically stacked layers when 1 µm of Parylene C is used as a spacer to separate the metal layers[47]. For SU-8 based architecture, Luan et al. only used a 500-nm-thick dielectric layer to ensure reducing the capacitive coupling from between metal traces and the surrounding medium of the probe to less than 1%[48]. Similar multi-layer metal processes can also be adopted for long-aspect ratio stainless steel neural probes. However, the longer shank for non-human primate probes necessitates careful design of the insulation layers in the vertical stack to minimize crosstalk between metal trace layers. To demonstrate the feasibility of this approach for implementing high-density neural probes, we have demon-strated double-layer trace lithography to implement a 32-channel device by stacking 16 channels in each metal layer (Pt/Au/Pt stack or Cr/Au/Pt stack). We have also demonstrated 32 channel steel-trode with single layer of metallic traces. Therefore, it is feasible to vertically stack additional layers with 32 channels to further scale up the channel count and density of the steeltrode.

### Steeltrode fabrication process is compatible with different flexible polymer insulation materials

Proper choice of insulation material to encapsulate the con-ductive traces is of paramount importance to ensure long-term stable recording performance of the implantable electrical recording neural interface. Firstly, the chosen material has to be biocompatible to ascertain minimal foreign body reaction from tissues when the device is implanted[49]. Moreover, the material must be biostable, meaning that the material should not corrode or delaminate during the time when the probe remains implanted[49]. In addition to being biocompatible and biostable, these materials need to withstand prolonged exposure to the biological environment and retain their insulation properties. There are multiple insulation materials that can be used in the design of steeltrodes. Dielectrics such as silicon dioxide, silicon nitride or semiconductors such as silicon carbide, as well as biocompatible polymers such as Polydimethylsiloxane (PDMS), Parylene C or Polyimide can be used. In this paper, we demon-strate designing steeltrodes with PDMS/Parylene C and SU-8 insulation materials. SU-8 is biocompatible and is a soft polymer[50]. It is widely used as the insulation material both in surface and also penetrating neural probes[48,51]. Moreover, because it is directly photodefinable, it significantly simplifies the fabrication process and is a great option for rapid prototyping and manufacturing of new designs of steeltrode devices, parti-cularly for acute in vivo testing.

To enable monolithic integration of flexible tether cables with steeltrodes, we have also used PDMS/Parylene C materials with much smaller flexural modulus values (compared to SU-8) to implement the insulation layer that can also serve as the material to realize a flexible cable[52]. In this design, PDMS serves as a planarization layer to reduce the surface roughness of stainless steel to facilitate high-resolution lithography. Parylene C layer deposited on the PDMS layer encapsu-lates the metallic interconnects and works as the primary insulation for the neural probe. Both Parylene C and PDMS are highly biocompatible polymers and have been used in FDA-approved medical implants[53]. Parylene C is also compatible with high-resolution lithographic pro-cesses to pattern sub-micron features[54,55]. Moreover, Parylene C has significantly low moisture absorption rate among commonly used polymer insulations for neural probes[49,56]. Such good water barrier

properties are essential for ensuring consistent signal quality during chronic electrical recording. Considering these properties, PDMS/Parylene C was chosen to implement our steeltrode architecture with a flexible tether cable for potential long-term recording.

## Results

### High resolution neuronal recording from large brains using steeltrodes

Our large-animal steeltrodes feature up to 8-cm-long stainless steel shanks with a compact small cross-section of 140 µm × 280 µm. Stain-less steel has high resistance to fracture, making our implant mechanically robust even at such an extreme form factor. We have demonstrated probes with 16–100 electrical recording electrodes on a single shank, with much larger than the channel density of commer-cially available stainless steel neural probes for large animals[30]. Such high-density electrodes can be implemented on both fully rigid shanks or on hybrid rigid-flex shanks, where the implantable rigid stainless steel shank is attached to the backend circuitry via highly flexible tether cables. Figure 1h shows SEM image of the tip of a steeltrode showing its smooth sidewalls, and 15-µm-diameter microelectrodes with a pitch of 50 micron. Images of fully rigid and hybrid rigid-flex steeltrodes are shown in Fig. 1b, c, respectively. This flexible cable in the rigid-flex design allows the implantable portion of the probe to move independently of the backend part that accommodates elec-trical connectors and circuitry. As a result, once inserted into the brain, the implanted shank of the stainless steel probe can move indepen-dently of the backend, reducing tissue damage and preserving signal quality by mitigating tethering forces from skull attachment during brain micromotions and sudden bodily movements[57,58]. This parti-cularly helpful for long-term neural recordings. Steeltrodes are pack-aged with flexible PCBs and zero-insertion-force (ZIF) connectors to ensure a compact footprint. Due to its small form-factor packaging and narrow cross section, multiple steeltrodes can potentially be implan-ted at different depths of the non-human primate brain for simulta-neous high-density neural recording from different brain regions (Fig. 1a). A long and robust implantable stainless steel shank enables access to deep parts of macaque brain. With 8-cm-long steeltrodes, we have demonstrated high-resolution electrophysiology recording from the auditory cortex of macaque brain (Fig. 1i). In summary, our novel neural interface utilizes the excellent mechanical properties of stain-less steel and the unmatched customizability of lithographically defined microscale features to enable distributed simultaneous recording of neural activity from deep regions of the large animal brain.

### High-resolution lithography on commercially available rough stainless steel using a planarization process

We have developed a fabrication process that allows high-density electrodes and electrical traces lithographically defined on flexible polymer layers monolithically integrated on the stainless steel sub-strate. This manufacturing method utilizes planar micro- and nano-fabrication processes on stainless steel substrates and mostly incor-porates parallel fabrication processes and can be scaled for mass production of such devices. Commercially available thin (thickness of 125 µm and lower) stainless steel sheets have high surface roughness. Topography and surface roughness of the stainless steel complicate high-resolution lithography on these substrates. It is possible to polish and smoothen the surface of thin stainless steel substrates. Such wafers are even commercially available (Valley Design Corp., Shirley, MA, USA). However, the polishing process can induce excessive stress in the thin film, which can lead to severely bowed wafers which are not suitable for planar fabrication processes. Therefore, a surface planar-ization process is essential to achieve high-resolution lithography on commercially available low-stress stainless steel substrates (SS 316, McMasterCarr), which have an inherent surface roughness with a

coarse texture. Conformal deposition of a thin-film polymer insulation material is not sufficient to planarize the surface roughness of stainless steel. For example, 7 μm Parylene C, deposited via a chemical vapor deposition process, only marginally reduced the surface roughness. Due to this roughness, lithographic resolution on Parylene C on stainless steel was therefore limited to ~2.5 μm. Whereas spin-coating SU-8 2010 of the same thickness (7 μm) proved to be much more effective in planarization; features with dimensions down to ~1 μm can be patterned on stainless steel (Fig. 2a). SU-8-based planarization process was used for realizing steeltrodes with fully rigid shanks (details discussed in Methods section). In addition to planarization, SU-8 functions as the top and bottom insulation layer for the neural probe. Moreover, since SU-8 is photodefinable, both the device outlines and the electrode openings can be patterned directly with photolithography in SU-8, without the need for plasma-based dry etching processes. Dry etching processes used for Parylene C, PDMS, and Polyimide (non-photodefinable) can lead to oxidization of the Platinum electrode surface that necessitates an additional electrochemical cleaning process or a physical etching process to restore the electrode impedance to its typical values[59,60]. Therefore, using SU-8 insulation not only contributes to expedite the fabrication process for rapid prototyping of different variants of rigid steeltrodes, but also mitigates the need for post-fabrication treatments of the electrode surfaces. Figure 2b shows a simplified fabrication process flow for realizing steeltrodes with SU-8 insulation.

To ensure the flexibility of tether cable for rigid-flex steeltrodes, the insulation material needs to have low flexural modulus. Compared to SU-8, Parylene C has almost 1000 times lower flexural modulus and as a result, its mechanical properties are better suited for steeltrodes with flexible tether cables. However, as shown in Fig. 2a, Parylene C itself is inadequate to planarize the stainless steel substrate to enable high-resolution lithography. To address this issue, we have developed a process for planarizing stainless steel with a combination of PDMS and Parylene C. PDMS has even lower Young's modulus compared to Parylene C and is suitable for realizing integrated flexible tether cables. However, PDMS is porous and has a high level of permeability, which makes it unsuitable as an insulation layer[5]. Moreover, its low surface energy leads to poor adhesion to photoresist, making photolithography directly on PDMS layer problematic[61]. To overcome these issues, a 7-μm-thick conformal layer of Parylene C is deposited on the PDMS layer at room temperature. Parlyene C layer combined with the PDMS layer forms the bottom insulation layer for the probe and functionalizes the PDMS surface for contact photolithography. As shown in Fig. 2a, a bilayer stack of PDMS–Parylene C provides sufficient planarization to enable high-resolution lithography (~1 μm features) to manufacture high-channel density steeltrodes with flexible tether cables for potential long-term recording in different animal models. With a 1 μm space and trace, the crosstalk between adjacent traces over the entire length of the probe is minimal, as shown in the previously reported crosstalk analyses[44]. Additional fabrication details are provided in the Methods section, with process steps shown in Supplementary Fig. S2.

### Multilayer lithography on stainless steel substrate facilitates large number of channels while retaining a small cross section of the probe

We can leverage multilayer metal trace fabrication on stainless steel to drastically increase the channel count of the neural probe without increasing the shank width. Such multilayer processes have been demonstrated for flexible probes with Parylene C[62,63] and SU-8[48] insulation. On Stainless steel substrate, we have demonstrated the feasibility of multilayer process with SU-8 insulation. Figure 2c illustrates a simplified microfabrication process flow diagram using which we have implemented a multilayer steeltrode on SU-8 with two vertically stacked metal layers (Pt/Au/Pt or Cr/Au/Pt stacks). Figure 2e, f shows micrographs of vertically stacked metal layers and the microelectrodes realized on stainless steel with SU-8 insulation and intermediate layers. We have demonstrated a 32-channel stainless steel probe implemented by vertically stacking two layers of metal traces separated by 2.5-μm-thick SU-8. In both layers, metal traces are 4 μm wide with 4 μm pitch, similar to the 16-channel variant of the probe with a single metal layer. Using this multilayer stacking of metal traces, we can scale up the number of channels significantly with minimal increase of the shank thickness. Moreover, this approach obviates the need for even narrower interconnect features using higher resolution lithographic techniques (i.e., deep-UV lithography, or electron-beam lithography) to retain the cost-effectiveness and scalability of the manufacturing process. When using narrower interconnects, especially in multi-layer designs, the fabrication process has to be optimized to achieve a high yield.

### Highly customizable, high-density electrode arrangements on steeltrode enabled by high resolution single and multi-layer lithography

Taking advantage of photolithography-based microfabrication process that we have developed on stainless steel, we could realize steeltrodes with highly customizable recording electrodes with variable channel densities and electrode arrangements. We have fabricated devices with 16 and 32 channels with inter-electrode separation as small as 25 μm. Different versions of these probes with different electrode arrangements have also been realized. Figure 1e–g shows microscope images of steeltrodes with linear, edge and honeycomb arrangements. Such devices can enable systematic study of the effect of recording electrode sizes, densities, and arrangements in terms of single-unit isolation yield as well as temporal and spatial frequencies in neural activities in the primate brain.

Apart from high degree of customizability and channel density, our proposed microfabrication platform also facilitates realization of high-channel density robust neural probes on stainless steel. The 16 and 32 channel devices that we have demonstrated utilize conservative metal traces with 2.5 μm width. However, as shown in Fig. 2a, we can pattern metallic features with much smaller dimensions on stainless steel. Using a conventional contact photolithography process (at the wavelength of 320 nm, Karl Suss MA6), we have demonstrated line-widths of 1 μm with 1-μm inter-trace space on stainless steel substrate. This critical dimension enables fabrication of probes with ~100 channels on a narrow shank spanning 250 μm, using only a single layer of metallic traces. Moreover, by leveraging the multilayer fabrication process discussed in the last section, we can massively scale up the number of channels. For example, using a 4-layer metallic trace lithography, we can accommodate ~400 channels on the same 250-μm-wide shank with an active recording span of 20 mm (assuming linearly arranged microelectrodes with 50 μm pitch). Therefore, we can realize steeltrodes with substantially larger channel counts compared to commercially available stainless steel neural probes using the conventional lithography process. It is worth noting that there is potential to further increase the channel count and density by employing advanced lithography techniques such as deep-UV lithography or electron-beam lithography to pattern sub-micrometer wide metallic traces on polymer layers[43,55]. However, it's important to acknowledge that these advanced lithographic processes are sequential in nature compared to conventional contact lithography. Consequently, the increase in channel count and density comes at the expense of manufacturing scalability. In contrast, our utilization of conventional photolithography processes ensures the fabrication of steeltrodes with substantially higher channel counts and density while retaining the crucial aspects of manufacturing scalability for high-throughput production.

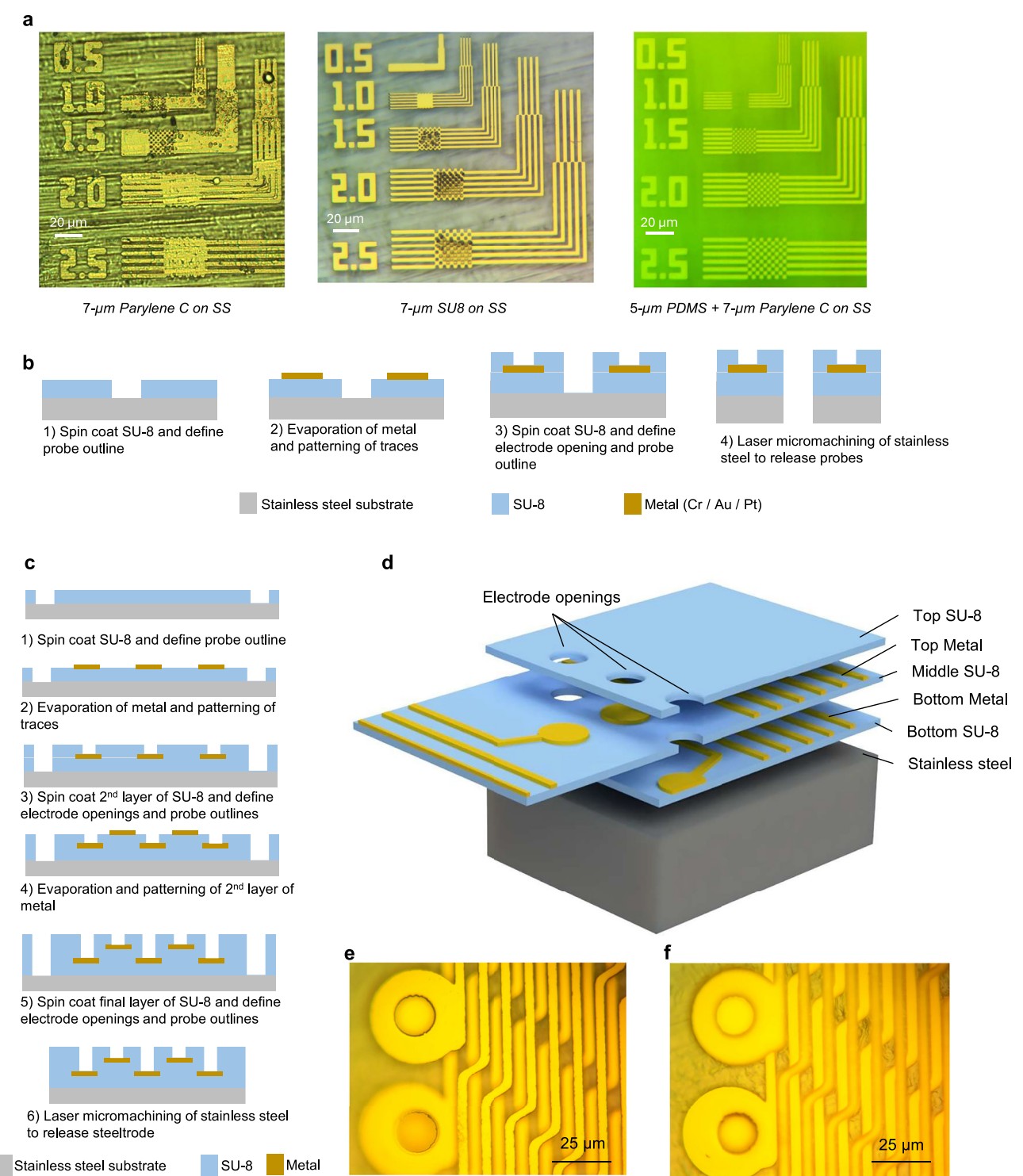

**Fig. 2 | Microfabrication of steeltrode on a commercially available stainless steel substrate. a** Effect of SU-8 and PDMS–Parylene C-based planarization on lithography resolution on Stainless steel (SS) substrate. **b** Process flow for fabrication of steeltrode with SU-8 insulation. **c** Process flow for fabrication of steeltrode with bi-layer metal traces. **d** Schematic cross section showing different layers of steeltrode with bi-layer metallic traces insulated in SU-8. Magnified micrographs showing features of top metal layer and bottom metal layer are shown in (**e**) and (**f**), respectively. Images are representative of devices fabricated through the microfabrication process shown in (**c**), in which 75 devices were produced with consistent structural features.

## High-throughput laser micromachining process for precise singulation of individual steeltrodes

High-density metal traces are insulated using a top polymer insulation layer, while selectively exposing the recording electrodes, backend contacts and the probe outlines via lithographic processes (details in Methods section). In the final step of the fabrication process, the stainless steel probes were released from the stainless steel wafer.

Following wafer-level microfabrication, we employ a customized laser micromachining process (Micron Laser Technology, Inc., Hillsboro, OR, USA) to release the steeltrode devices with long implantable shanks from the stainless steel substrate. A 200 W fiber laser (kerf: 20 µm, 300 PSI air assist) was utilized to ablate the stainless steel around the probe outlines to release each of the probes from the wafer. Using a combination of computer-aided-design (CAD) file and

lithographically defined alignment marks on the stainless steel wafer, the laser's position and movements were precisely controlled within ±10 μm. This process allows customization of the tip profile (shape, opening angle, etc.) through CAD specification. Supplementary Fig. S3a shows the laser etching trajectory used to singulate a steeltrode on a processed stainless steel wafer, along with representative images (Supplementary Fig. S3b) of the steeltrodes after their release via laser micromachining.

Adapting commonly used planar microfabrication processes for stainless steel, followed by precision laser micromachining makes our neural interface platform promising for highly scalable wafer-scale fabrication.

## Monolithic flexible tethering cable is realized by selective electrochemical etching of stainless steel

Brain micromotions is reported to be one of the factors causing tissue response during chronic recording. Rigid probes tethered to the skull can impart force on the brain tissue due to the relative motion of brain with respect to skull. Flexible tether cables on probes allow rigid neural probe to move with brain micromotion and minimize tethering forces. In our proposed hybrid stainless steel-polymer platform, we have demonstrated post-fabrication processing to manufacture probes with highly flexible polymer tether cables that separate the rigid implantable portion of the probe from the backend ZIF connector that connects to the recording interfaces. This process is based on electrochemical etching (details available in Methods section) of some portion of stainless steel shank of the microfabricated rigid stainless steel probe insulated with PDMS–Parylene C. Steeltrode variant with PDMS–Parylene C is preferred over SU-8 for implementing tether cables since SU-8 has few orders of magnitude higher flexural modulus compared to Parylene C. Therefore, compared to SU-8, tether cables made from Parylene/PDMS are significantly more flexible. Figure 1c, d shows micrographs of steeltrode with PDMS/Parylene C tether cable.

## Rigid stainless steel backend allows fast, efficient, and reusable ZIF connectors for interfacing with the recording electronics

Packaging of high-channel-count neural implants with the recording amplifier circuitry is challenging. Wirebonding, one of the most widely used packaging processes, is time-consuming and requires professional expertise to achieve high yield when bonding to high-density contacts to minimize the form factor of the probe backend. It also requires epoxy encapsulation to protect the fragile connections. Our stainless steel probes are interfaced with a custom-designed flexible printed circuit board (fPCB) via a compact, low-pitch zero insertion force (ZIF) connector (Hirose FH26 Series). The backend bondpads of our probes are designed to complement the contacts of the ZIF connector. Stainless steel support layer underneath the bondpads of the backend of the probes makes it convenient for plugging the probes directly into the ZIF connector sockets without any additional stiffener layers. Moreover, the probes can be easily unplugged from the connectors and the same ZIF connector and PCB assembly can be reused multiple times with different probes. The simplified packaging process enabled by the stainless steel backend provides effortless, hassle-free, and swappable interface to recording circuits with high yield while maintaining a compact profile of the probe backend.

## Electrodeposition of conductive polymers on electrode surface to enable high-fidelity neural recording

Electrochemical impedance of recording electrodes is an important criterion that determines the extracellular recording quality and the background noise. We have characterized the electrochemical impedance of 15-μm-diameter Pt electrodes on steeltrodes in 1X phosphate-buffered saline (PBS). At 1 kHz frequency, the electrodes yield an impedance of $1.07 \pm 0.093$ MΩ ($N = 14$ channels) in PBS solution (Supplementary Fig. S4a). Electrochemical impedance of <2 M-ohms has

been reported to be sufficient for single unit isolation even with commercial neural amplifiers with relatively low input impedances[64]. However, lower impedances are desired to reduce thermal and biological noises during recording. Different materials such as Pt grass, Pt black, Au, PEDOT:PSS, and PEDOT:CNT are often deposited post-fabrication to reduce electrode impedances, while keeping the electrode size unchanged[64–68]. For our stainless steel probe, we electrodeposited conductive polymer PEDOT: PSS on the bare Pt electrodes (details in Methods). As a result, PEDOT:PSS coated electrodes ($N = 14$) on steeltrodes exhibited impedances of $51.2 \pm 2.26$ kΩ at 1 kHz (Supplementary Fig. S4a), representing a ~40 fold reduction compared to bare Pt electrodes.

## Bending experiments validate the mechanical durability and electrical performance of steeltrodes

A key requirement for long-aspect ratio intracortical neural probes, particularly those intended for repeated use or for applications involving challenging implantation scenarios in non-human primates, is the ability to withstand significant mechanical stress without fracturing. While conventional silicon-based probes are prone to brittle failure, steeltrode's stainless steel substrate provides a much higher fracture toughness, thereby mitigating concerns of breakage during or after insertion. To experimentally quantify this mechanical robustness, we performed bending tests on fully fabricated steeltrode devices and monitored their electrical functionality before and after each deformation.

In our experiments, a 3D-printed mount was used to secure the backend of the steeltrode, while a motorized indenter (driven by a linear actuator, controlled via a microcontroller) applied a bending force to the tip of the probe (Fig. 3a). The tip was displaced from its neutral (undeflected) position by 10, 20, 30, and 40 mm, respectively, and was then returned to the neutral position. Figure 3b illustrates three representative displacement levels of the probe tip under this controlled bending load. The steeltrode used for this experiment was 78 mm long. After each displacement step, the steeltrode was immersed in 1X phosphate-buffered saline (PBS), and the electrochemical impedance of all microelectrode channels was measured using the same three-electrode configuration described previously. As shown in Fig. 3c, d, the electrochemical impedance magnitude ($1.55 \pm 0.13$ MΩ at 1 kHz) and phase ($-70 \pm 1$ deg at 1 kHz) of 25 recording electrodes remained virtually unchanged relative to baseline even after the maximum applied deflection of 40 mm.

These observations confirm that the monolithically microfabricated stainless steel probe and patterned electrode layers are highly resilient to large bending forces. Consequently, the electrical functionality of the steeltrode remained stable despite being repeatedly subjected to substantial mechanical stress that would likely fracture silicon-based probes with similar dimensions. This robust mechanical performance demonstrates steeltrode's suitability for neural interfacing in scenarios where significant bending or incidental loading may occur during implantation, explantation, or handling.

## Single-unit and multi-unit auditory-evoked potential recorded in macaque brain using steeltrodes

The functionalities of the probes were tested in adult *Macaca mulatta* by recording both spontaneous and stimulus-locked evoked response from auditory cortex. Our goals were to show (i) that the steeltrodes are compatible with standard recording setups used for acute recordings in NHPs, (ii) that the probes were long enough to reach deep brain structures such as auditory cortex, and (iii) that the quality of the neural signals was good enough for detecting single and multi-unit activity, as well as laminar local field potentials, current-source densities and frequency following responses.

The non-human primate experiments were conducted on an awake head-fixed macaque in a sound-attenuated recording booth.

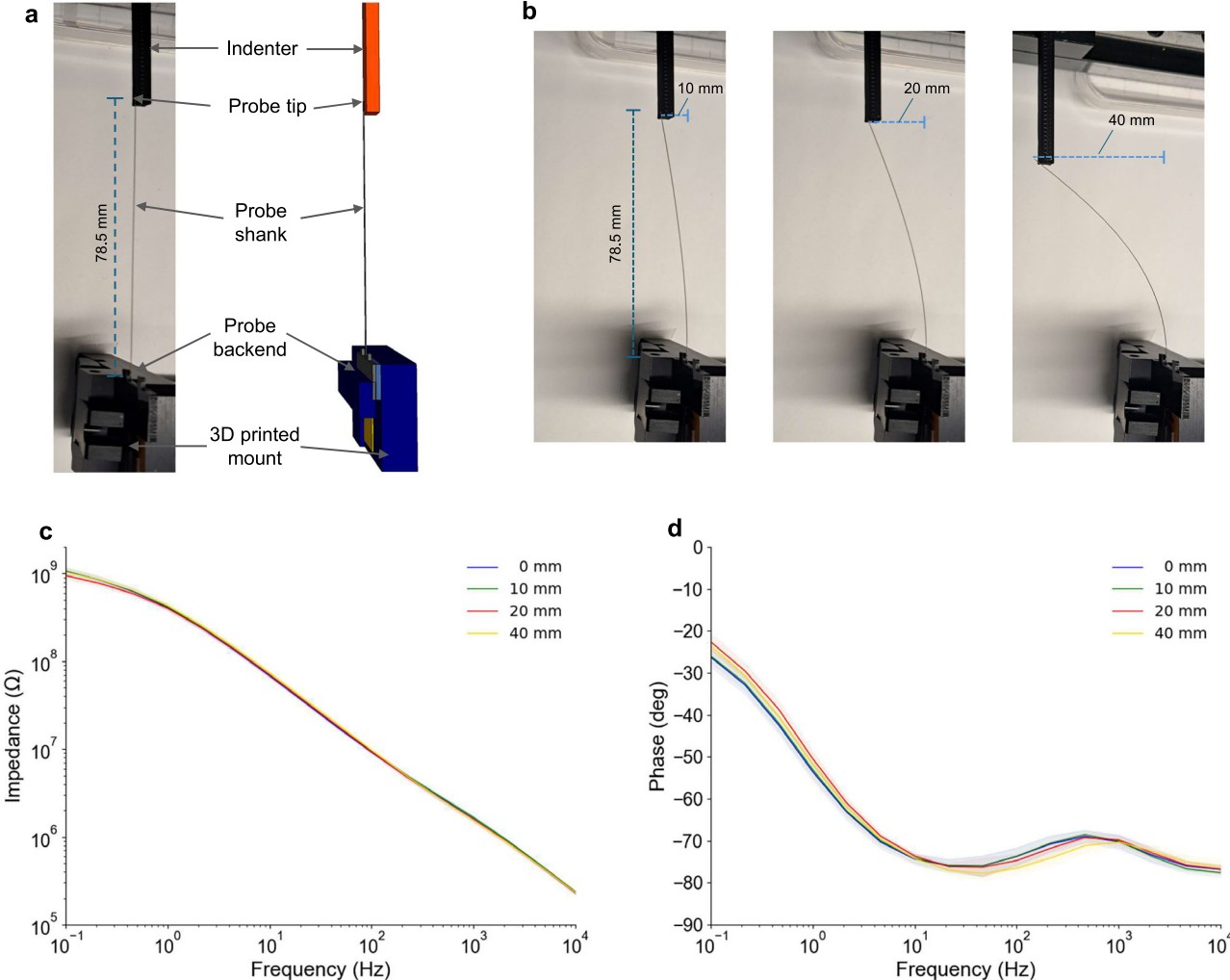

**Fig. 3 | Mechanical bending tests of the steeltrode. a** Photographs and schematic diagram of the experimental setup, showing the steeltrode clamped in a 3D-printed mount, where a motorized indenter is placed next to the probe tip. **b** Steeltrode is bent at the tip by 10, 20, and 40 mm using the indenter. **c**, **d** Electrochemical impedance magnitude and phase plots, respectively, after 0, 10, 20, and 40 mm of deflection, revealing minimal changes across frequencies from 0.1 to $10^4$ Hz. Data are presented as mean values ± SD across $N = 25$ measurements. These results confirm that the steeltrode's electrodes remain electrically stable despite substantial bending.

Steeltrode was implanted in the brain through an implantation grid (targeting auditory cortex) in a recording chamber placed over frontal cortex. Prior to steeltrode insertion, the dura was punctured using a stainless steel guide tube. The steeltrode was connected to a neural amplifier for neural data acquisition. Details of the implantation process are available in the Method section.

### Spontaneous activity recording from auditory cortex

We recorded spontaneous neural activity at multiple depths corresponding to different layers of auditory cortex using a 16 channel steeltrode with high-density microelectrodes with 25-μm inter-electrode separation. To confirm accurate positioning of the steeltrode in the auditory cortex, we relied on identifying responses evoked by sound, a method detailed in the following section. We used an in-house neural data analysis pipeline based on Mountainsort[69] to isolate high-fidelity single and multi-unit activities from the raw neural recording data. The raw data were first band-pass filtered between 500 and 7500 Hz. Spikes were then identified by setting a threshold of 4 times the standard deviation of noise for each channel. Figure 4c shows the high-pass filtered signal from 15 functional recording channels, with threshold crossing events identified. Putative units were clustered using principal component analysis (PCA). Figure 4d shows average

waveforms across laminar recording channels for 2 putative units identified from spike sorting. Inter-spike interval (ISI) histograms are also shown for these 2 units. For Unit 1, we note that almost 2% of the spiking events occur within the typical neuronal refractory period of 2.5 ms. This suggests that Unit 1 corresponds to a multi-unit response, as multiple neurons may be contributing to the spiking activity. For Unit 2, none of the spiking events violate the refractory period threshold, suggesting that it is a single unit.

### Identifying recording locations in auditory cortex

To ensure precise placement of the steeltrode in the auditory cortex, a passive white-noise burst paradigm was employed. This protocol used 25-ms-long white noise bursts with 5-ms-long rise and fall time at 5 different intensities (46, 56, 66, 76, 86 dB SPL (sound pressure level)). These sounds were presented every 400 ms, with an additional ±50 ms of jitter to introduce variability (as illustrated in Fig. 5a). Each session of the auditory stimulation protocol comprised 250 trials. The flat frequency spectrum and the fast rise time ensured strong responses across the entire tonotopic map of auditory cortex.

We employed current source density (CSD) analysis on the local field potential (LFP) signals to identify the specific neural generators responsive to auditory stimulation. This analysis played a crucial role in

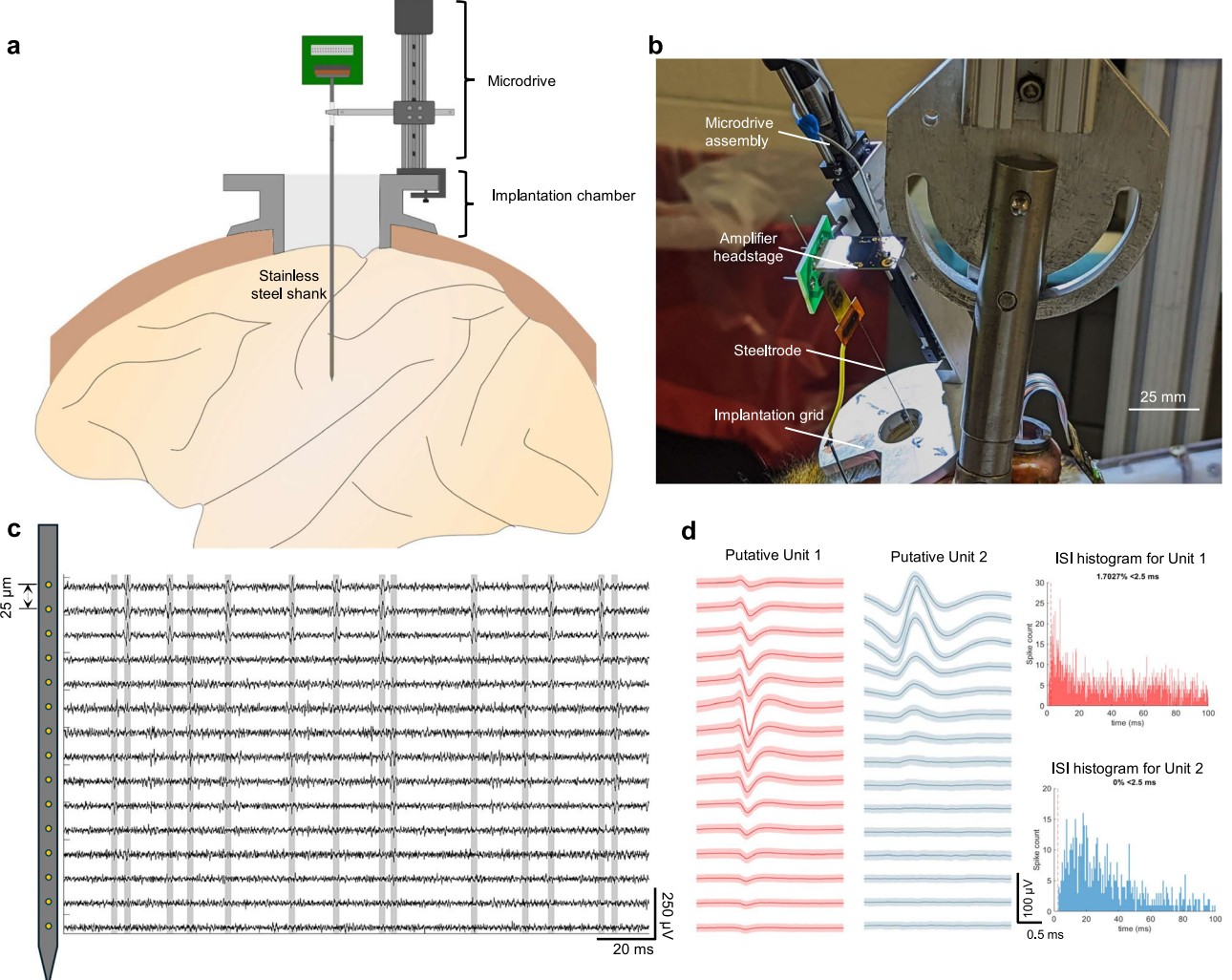

**Fig. 4 | Neural signal recording from macaque brain using steeltrodes.**
**a** Schematic of long-aspect ratio fully rigid steeltrode implantation in macaque brain. **b** Photograph of steeltrode implantation in macaque brain. Steeltrode packaged with flexible PCB is attached to a Microdrive which inserts the probe through a guide tube and implantation grid. **c** High-pass filtered neural data recorded with steeltrode (with 25 micron inter-electrode pitch) from macaque auditory cortex. Threshold crossing events corresponding to putative spikes are highlighted. **d** Spike sorted waveforms and corresponding ISI histogram plots for 2 different putative units. Data are presented as mean waveforms with error bands indicating ±SD.

determining the accurate placement of the steeltrode within the auditory cortex. Additionally, we analyzed sound-evoked multi-unit activity. Although an evoked response in the multi-unit band may suggest activity in fibers of passage of stimulus-responsive neurons, the presence of neural generator features identified through CSD analysis provides a more robust and reliable indication of local synaptic response within the auditory cortex[70].

To analyze the local field potentials (LFP), we first applied a low-pass filter with a cutoff frequency of 300 Hz to the raw neural signal. Next, using synchronization triggers presented at the sound onset, we segmented the filtered signals into 300 ms long epochs, ranging from −50 ms to +250 ms around the tone onset. Each electrode's epochs were then averaged over multiple trials after subtracting the common average response from all recording channels. The averaged evoked LFP responses across the laminar electrodes were visualized in Fig. 5b. To compute the CSD, we used trial-averaged LFP signals from five adjacent electrodes to calculate the hamming window smoothened second-order spatial derivative[71,72]. Figure 5c displays CSD features, revealing post-synaptic response of an ensemble of auditory responsive pyramidal cells near the tip of the probe as indicated by dipole-like neural features located near the bottom part of steeltrode.

To extract multi-unit activity (MUA), we first band-pass filtered the raw neural data in the range of 300−7500 Hz. Afterward, we applied full wave-rectification and a low-pass filter with a cutoff of 100 Hz to obtain the envelope of power in the frequency band of spiking activity. We computed the trial-averaged MUA for each channel by averaging the responses from epochs around the onset of auditory stimuli. Figure 5d depicts the auditory-evoked average MUA profiles across the cortical depth.

These analyses were performed online as the probe was advanced in small steps of 500−2000 μm. We considered the probe to have reached auditory cortex if (i) we identified a polarity inversion of the first LFP deflection that peaks at around 25 ms after sound onset and (ii) we identified robust sound-evoked multi-unit activity with a peak latency on the order of 20 ms around and below the depth of polarity inversion.

We also analyzed single and multi-unit sound-evoked activity. Figure 5e, f display the average multichannel waveforms, peri-stimulus time histogram, and raster plot for a putative unit whose firing rate exhibits a strong correlation with the auditory stimulation. Notably, we observed a significant increase in firing rate ~20 ms after the tone onset, which also aligns with peak latency of trial-averaged multi-unit activity.

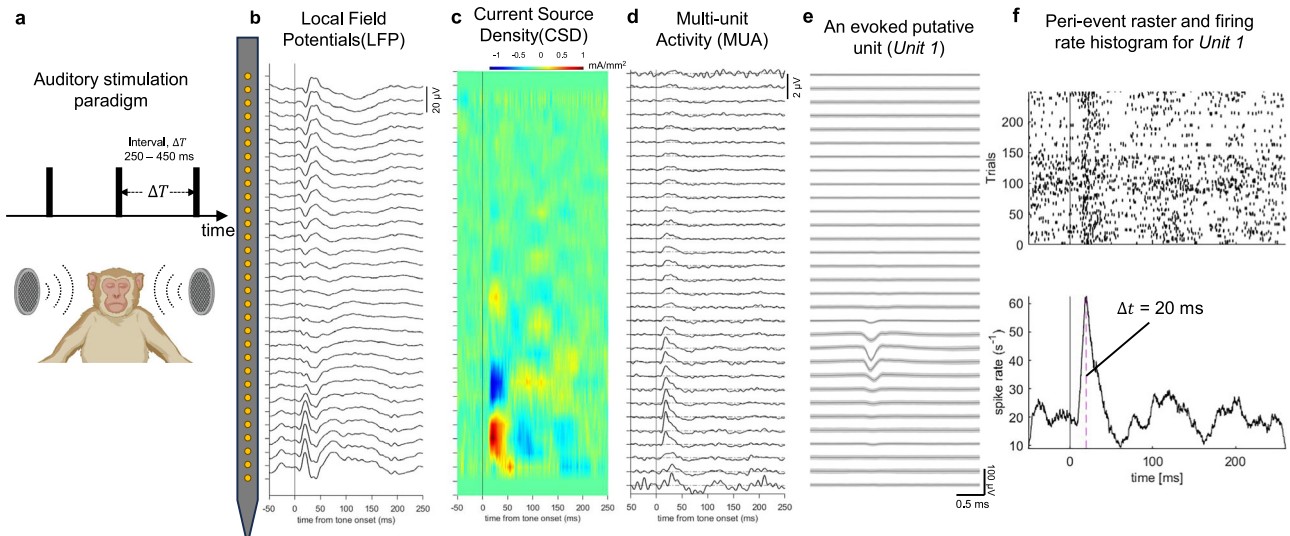

**Fig. 5 | Macaque auditory-evoked response measurement with steeltrode.**
**a** Auditory stimulation paradigm consisting of white noise bursts (created in BioRender. Chamanzar, M. (2025) https://BioRender.com/nznmfl3). There are 250 trials in each stimulation session that has white noise pulses occurring at 400 ± 50 ms intervals. In **b**–**d**, we are showing trial-averaged local field potentials,
current source density, and multi-unit band activity, respectively for the laminar recording. **e** Average waveforms (with error bands indicating ±SD) of a putative unit (Unit 1) that is responsive to auditory stimulus. **f** Peri-stimulus raster and firing rate histogram for Unit 1 showing significant increase of firing rate 20 ms after the stimulation tone onset.

## Tonal response field (TRF)

Tonal response field (TRF) refers to the patterns of evoked neural response in auditory cortex in response to sounds of different frequency and intensity. By applying a range of frequencies with different intensities, TRF can be constructed for different regions of auditory cortex. It is useful in understanding how the brain encodes and processes different properties of sounds, i.e., intensity, frequency and duration, etc. TRF is also used in the design and optimization of auditory prosthesis like cochlear implants, as well as for assessing damage after insertion of such implants[73].

In our experiments using steeltrodes, TRFs were measured with a standard receptive mapping task. Animal passively listened to 50-ms pure-tone pips with a 5-ms on/off taper (Fig. 6a). Tones were presented at 32 frequencies that are equidistantly spaced in $\log_2$ space between 90 Hz and 25 kHz, and at five intensity levels (20, 30, 40, 50, and 60 dB SPL). The interval between tones varied randomly between 0.8 and 1.2 s. Each frequency–intensity combination was repeated 10 times, resulting in a ~40-min recording session.

Auditory-evoked, trial-averaged multi-unit activity (MUA) responses were computed using the same approach described in our previous stimulation paradigm. Figure 6b presents the laminar trial-averaged tone-evoked MUA across cortical depth for a 7.1 kHz stimulus at 60 dB SPL. For each channel, the magnitude of the MUA response was determined by calculating the signal power within a specific temporal window—from 15 ms to 50 ms after the tone onset.

In Fig. 6c, a line plot shows the normalized MUA response as a function of stimulus frequency at three different intensities, specifically for the channel that exhibited the strongest tone-evoked response (indicated by the red arrow in Fig. 6b). The data reveal that the neuronal population at that location is strongly tuned to frequencies around 7.1 kHz. This preferred frequency is also known as best frequency in TRF literature. Selectivity to auditory stimulus of certain frequency is a hallmark of neurons in non-human primate auditory cortex. This frequency-sensitive response also indicates the successful placement of the steeltrode within the macaque auditory cortex.

Using the full set of MUA responses across all stimulus frequencies and intensities, we calculated tonal response field (TRF) map

for the most responsive channel in a specific recording session. Figure 6d displays this TRF map, illustrating how the response varies with tone frequency and stimulus intensity, and further highlighting the strong tuning at 7.1 kHz. The best frequency, identified as the frequency for which the evoked neural response is the strongest, varies at different regions of auditory cortex. TRF maps from two additional recording sessions—where the steeltrode was implanted at different locations in auditory cortex—are shown in Fig. 6e, f. These sites were located ~4.5 mm posterior, and 6.0 mm posterior with an additional 1.5 mm lateral offset, respectively, relative to the location corresponding to the TRF map in Fig. 6d. These spatial offsets were calculated based on the known 1.5 mm pitch of the cortical positioning grid. To enable direct comparison across sessions, each electrode's MUA response was normalized by the maximum response observed across all frequencies, intensities, and electrodes within that session. This yielded a normalized scale from 0 to 1, allowing standardized comparisons across recordings. The resulting TRF maps reveal distinct best frequencies of 17.4 kHz and 20.5 kHz at the two more posterior sites (Fig. 6e, f, respectively), reflecting regional variability in frequency tuning along the auditory cortex. Recordings were made at depths estimated to correspond to cortical layer 3, based on microdrive depth tracking, known cortical thickness, and characteristic tone-evoked MUA profiles.

Given the tonotopic organization of auditory cortex and the orthogonal insertion angle, adjacent contacts on the same shaft during each recording session showed very similar best frequencies; larger frequency gradients would emerge if steeltrode was inserted parallel to the superior temporal plane from posterior to anterior. Although we observed uniform best-frequency tuning across contacts at a single implantation site, future studies can leverage the steeltrode's high-density array to resolve millisecond-scale response-latency differences among neurons that may share similar best frequencies. Additionally, this fine spacing will facilitate probing of local synaptic interactions, which are strongest among closely spaced cells. The successful acquisition of TRF maps using steeltrode demonstrates its effectiveness and strong potential for probing functional organization in deep brain structures, offering valuable insights for neuroscientific research.

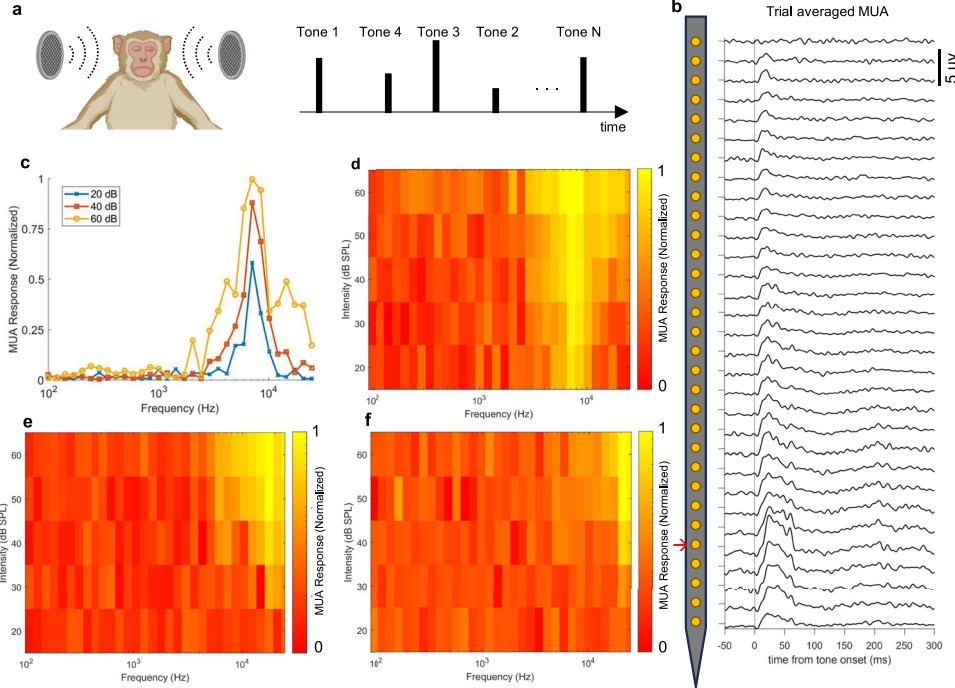

**Fig. 6 | Macaque tonal response field measurement with steeltrode.**
**a** Schematic (created in BioRender. Chamanzar, M. (2025) https://BioRender.com/nznmfl3) of the stimulation paradigm used to generate the tonal response field. Tones with varying frequencies and amplitudes were presented at randomized intervals. **b** Trial-averaged laminar multi-unit activity (MUA) recorded from the macaque auditory cortex in response to a 7.1 kHz tone at 60 dB SPL. **c** Normalized MUA response calculated from the trial-averaged data for the channel showing the strongest response (indicated by the red arrow in (**b**)). This panel demonstrates frequency tuning at three different stimulus volume levels across frequencies from 90 Hz to 25 kHz, corresponding to the tonal response field map shown in (**d**). Tonal response field maps from two additional locations in the macaque auditory cortex are shown in (**e**) and (**f**).

## Frequency following response (FFR) in auditory cortex

Recording yield/quality of steeltrode in the low frequency band were characterized by acquiring frequency following response from macaque auditory cortex. FFR is a neural response in auditory cortex that is synchronized with the phase of the auditory stimulus waveform. There are some inherent challenges associated with FFR detection. These challenges stem from FFR's characteristically low signal-to-noise ratio and also elusive nature of the specific locations of FFR neural generators which has been the source of continued scientific debates. We aim to demonstrate high-density steeltrode's capability in detecting such challenging signals and its utility in potentially providing new insights into triangulating FFR neural generators. FFR is a useful tool for understanding the auditory system and provides information about how sound and speech are encoded and processed in the brain. FFR is also useful in diagnosis of neurodevelopmental, speech and learning disorders[74,75].

FFR is characterized by neural response in the frequency band ranging from 70 to 500 Hz, which follows the temporal periodicity envelope of the auditory stimuli. The click train stimuli used in this study were specially designed to correspond with the range of fundamental frequencies (F0) found in a Mandarin tone, corresponding to the vowel /yi/[76]. It is typical for Mandarin words to be used in FFR paradigm, as it is a tonal language, which means that the pitch and tone of the word can change its meaning. The chosen mandarin tone had a time varying F0 contour that changed from 89 to 111 Hz. The stimulus click train (Fig. 7a) was synthesized from this tone to match the timing and amplitude characteristics of the fundamental frequency. Specifically, the F0 cycle timing was determined by the moment of peak pressure, and the intensity of the clicks was set at double the absolute peak amplitude[76]. The presentation of the stimuli was randomized, with varying inter-stimulus intervals ranging from 300 to 500 ms. During each recording session, which lasted 40 min, a total of 500

repetitions of each tone and polarity were presented, amounting to 4000 sweeps in total.

FFR from the recorded data was extracted by first band pass filtering the raw signal at each of the microelectrodes of steeltrode in the range of 70–500 Hz and then by averaging the time-locked epochs around the onset of each type of tone. Epochs spanning 350 ms around the tone were considered. Figure 7b shows the trial averaged FFR at a single channel of steeltrode corresponding to the stimulus click train.

Robust high-density electrodes on steeltrode can be a valuable tool to study the sources of neural generators for FFRs. Laminar distribution of FFR (Fig. 7c) can be calculated from the electrode array of steeltrode. Moreover, neural generators for FFR can be identified and isolated by calculating CSD across the cortical depth. In particular, high-density electrode on steeltrode, we can facilitate extraction of highly localized information regarding FFR sources that can be used to improve our understanding of speech encoding and auditory processing in primate and non-human primate brain.

To assess the impact of electrode spacing on the spatial accuracy in identifying neural generators of FFRs, we compared the CSD results from recording with three different electrode densities, i.e., 50 μm, 100 μm and 150 μm pitch. This involved synthesizing two datasets from the original raw data (recorded with 50 μm pitch): one using every other channel to achieve an effective pitch of 100 μm, and the other using every second channel for an effective pitch of 150 μm. From these laminar FFR data, CSD can be calculated by taking the second-order spatial derivative. Figure 7d shows the laminar CSD for FFR measurement using steeltrode with 50 μm pitch. As it can be seen, in the top half of the probe, we can observe two distinct dipole-like field arrangements corresponding to sinks and sources that can be attributed to local neural generators of FFR. However, with increased electrode spacing, there is a noticeable decrease in spatial resolution for identifying these neural generators, as dipole-like features spread out

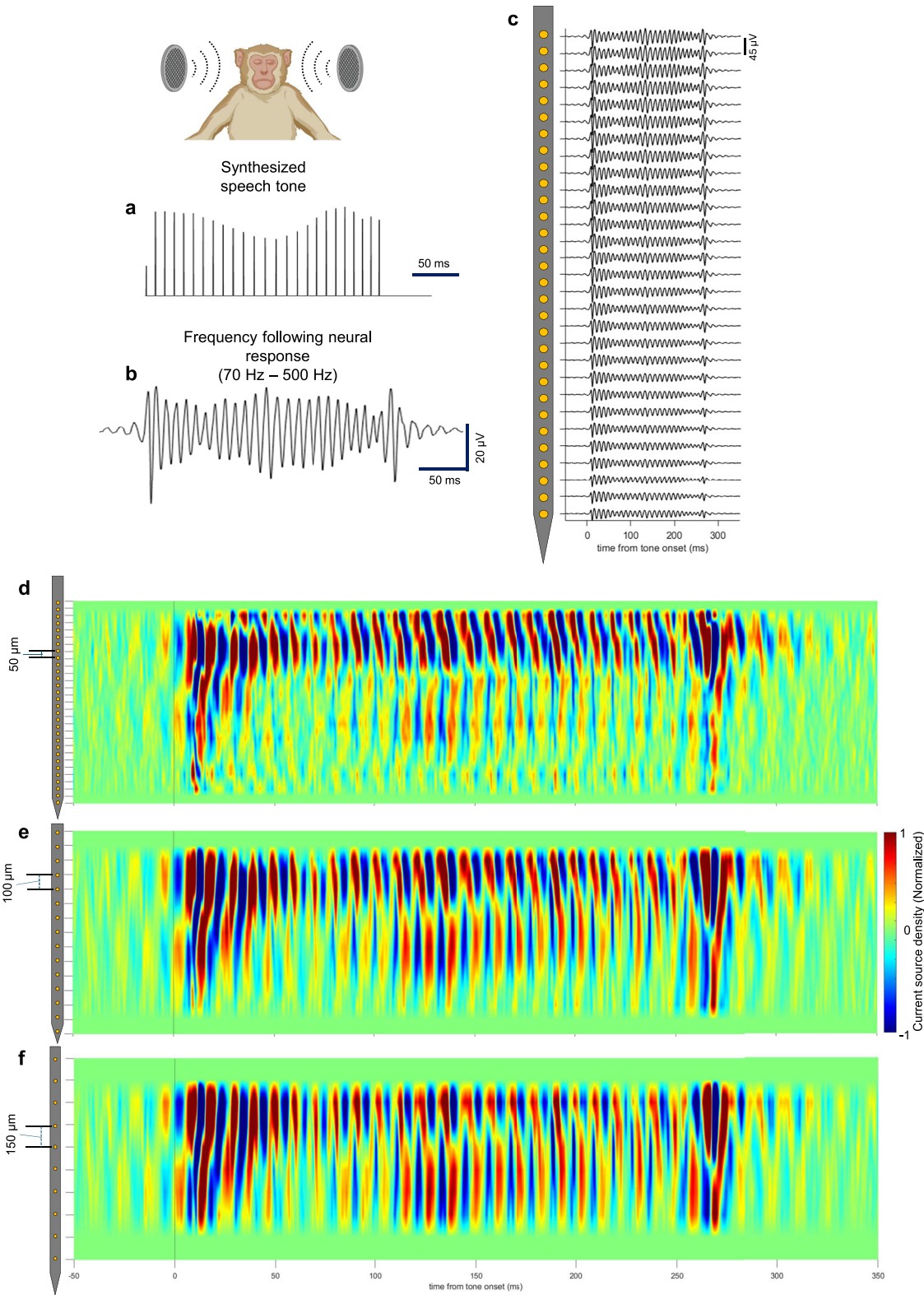

**Fig. 7 | Macaque frequency following response measurement with steeltrode.**
**a** Synthesized speech tone stimulus along with the experimental schematic (created in BioRender. Chamanzar, M. (2025) https://BioRender.com/nznmfl3). **b** The corresponding frequency following neural response. **c** Trial-averaged laminar frequency following response (70–500 Hz) measured with Steeltrode. **d–f** Current source density plots for three datasets with varying pitches of the electrodes (linear arrangement): 50 μm, 100 μm, and 150 μm, respectively. The local field potential data for 100 μm pitch (shown in **e**) and 150 μm (shown in **f**) were derived from the original data collected using a Steeltrode with a 50 μm pitch (shown in **d**). Current source density was then calculated from the synthesized data.

and lose definition (Fig. 7e, f). Particularly at a 150 μm pitch, crucial spatial information about upper neural generators is significantly diminished or lost. This underscores the utility of high-density steeltrodes for precise detection of local field potentials, current sources densities and shows their significance in advancing our understanding of auditory neuroscience and speech processing.

### Steeltrode with smaller shank is rigid enough to puncture rat dura and remains fully functional and reusable after repeated insertions

Using the highly customizable photolithographic patterning and laser micromachining process of our hybrid stainless steel polymer probe platform, we can also implement rodent steeltrodes with smaller shanks. We have fabricated 16- and 32-channel rodent probes with 1–2-cm-long stainless steel shanks. Much like the NHP probes with ultra-long shanks, these shorter rodent probes also benefit from mechanical robustness of stainless-steel substrate. In addition to being highly resistant to fracturing, these short steeltrodes are mechanically resilient enough to puncture rodent dura. Conventionally, for inserting silicon-based rodent probes, dura is surgically removed before the probe is implanted. However, this surgical process may lead to accidental tissue and vascular damage and tissue swelling. The mechanically superior material properties of stainless steel make the steeltrodes rigid and robust enough to withstand the force needed for penetration through rodent dura. Therefore, the probe itself can be used to puncture the dura and that precludes the need for a surgical procedure for dura removal. As a result, the risks associated with surgical removal of dura can be avoided.

We successfully demonstrated the efficacy of steeltrodes in puncturing the intact dura in a rat model. To achieve this, the probe was inserted into the rat brain at a controlled speed of 100 μm/s through the intact dura to a depth of 3.5 mm, targeting the somatosensory cortex and hippocampus. Following acute insertion and recovery, spontaneous neural activity was recorded, with the probe remaining in place for 2 h. Figure 8a shows different stages of steeltrode implantation through rat dura. To evaluate the benefits of dura piercing using the probe, a histological comparison (Fig. 8b) was conducted between probe-assisted dura penetration, probe insertion following surgical dura removal and no insertion. After electrophysiological recording, the animals were sacrificed, and their brains were fixed, sectioned and then processed using Nissl staining (NeuroTrace) to evaluate neural density in the superficial layers of the target site. For analysis, a region of interest (ROI) measuring 900 μm × 500 μm was chosen to consistently sample the superficial cortical layers and the expected local effect produced by dura removal or piercing, while limiting heterogeneity introduced by deeper layers or adjacent fields. This ROI size balances spatial coverage and measurement precision and supports robust biological replication. To account for inter-experimental variability, cellular densities from each biological replicate were divided by the highest control value and expressed on a normalized scale. This ensures that all reported values range from 0 to 100% relative to the maximal control response and allows direct comparison of relative reductions or increases across experimental conditions. Normalization to the maximum cell density of the control is appropriate here because the ROI area and sampling method were consistent across all groups, and the intent was to represent relative cellular preservation or loss rather than absolute density differences. Detailed procedures for in vivo rat experiments and histological analysis are provided in the Methods section. Histological examination revealed that dura removal resulted in a significant cell density reduction compared to the control case (-44.3% decrease versus control), whereas dura piercing caused only a modest reduction (-16.8% decrease compared to the control case). Cellular density in the dura removal case was also significantly lower compared to dura piercing (-33% decrease) as shown in Fig. 8c. These results suggest

that direct piercing of dura with steeltrode preserves more cells compared to surgical removal of dura and then inserting the steeltrode.

We also characterized the mechanical robustness and reusability of these rodent steeltrodes, highlighting their suitability for repeated use. We took a representative steeltrode (15 mm long) and carried out 10 successive dura penetrations in four rats. After every few insertions, we measured the electrochemical impedance of the electrodes on the probe (15 channels). Figure 8d, e compare the magnitude and phase of the electrochemical impedances at the baseline (before any implantation) and after 4, 8, and 10 insertions. Notably, minimal changes in electrode impedances were observed, indicating negligible degradation of the steeltrode integrity. Since the electrochemical impedance stability is closely related to the noise floor for in vivo recordings[64], this result implies that the steeltrode remains viable for high-fidelity neuronal recording after multiple penetrations. Furthermore, on the 10th implantation, we used the same steeltrode to record high-fidelity spiking activity in rat hippocampus across multiple channels. Representative average waveforms and the corresponding ISI histogram of putative units with amplitudes on the order of 100 μV are shown in Fig. 8h, i.

In summary, our rodent steeltrodes provide sufficient mechanical robustness for directly piercing through dura, which can minimize damage to superficial cortical layers by obviating the surgical removal of dura. Our experiments also demonstrate electrochemical stability of steeltrode's electrodes after multiple implantations through rodent dura, indicating its potential for reusability for recording high-fidelity neural signals.

## Discussion

The microfabricated high-aspect-ratio, high-density stainless steel neural probes introduced in this paper, enable high resolution electrophysiology recording from deep regions of non-human-primate brain both for detecting spontaneous and evoked activity. These robust neural probe implants can be used in a plethora of applications, ranging from basic science discovery to brain-machine interfaces with a path towards clinical applications in humans for inter- and intraoperative neural recording. The key engineering innovation that has enabled these great potential applications is the microfabrication process that we have developed for defining customizable high-density recording electrodes on long-aspect ratio stainless steel probe shanks. In this paper, we demonstrated neural probes with up to 8-cm-long implantable shank with a very small cross section of 140 μm × 280 μm. The 8-cm steeltrode easily enables reaching subcortical regions in macaques. In our experiments, the approach angle, chamber design and microdrive arrangement dictated that between 2 and 4 cm of the probe shank length remained out of the brain tissue, thus leading to an effective electrode length of 4–6 cm. Given that the dorso-ventral extent of the monkey brain is below 5 cm, the current setup would provide access to most of the brain, including most subcortical targets. A redesign of the setup can further increase the effective length to around 7 cm, thus providing access to the entire monkey brain. It is also possible to design longer probes for larger species. The length of the neural probe was limited by the size of the substrate (i.e., a 4-inch-diameter wafer) that we made for photolithography. Since we made this wafer out of large stainless steel sheets, the size of the wafer can be easily made larger (i.e., 6 inches or even 8 inches) to microfabricate neural probes, with implantable shanks of up to 18 cm, if needed. High resolution lithographically defined features on stainless steel are enabled by a planarization process using PDMS, Parylene C and SU-8. In addition to providing a smooth surface for lithography, these polymer layers function as insulations for the microelectrodes. Using selective electrochemical etching of stainless steel, we have created completely flexible tether cables attached to rigid shuttles and demonstrated hybrid rigid-

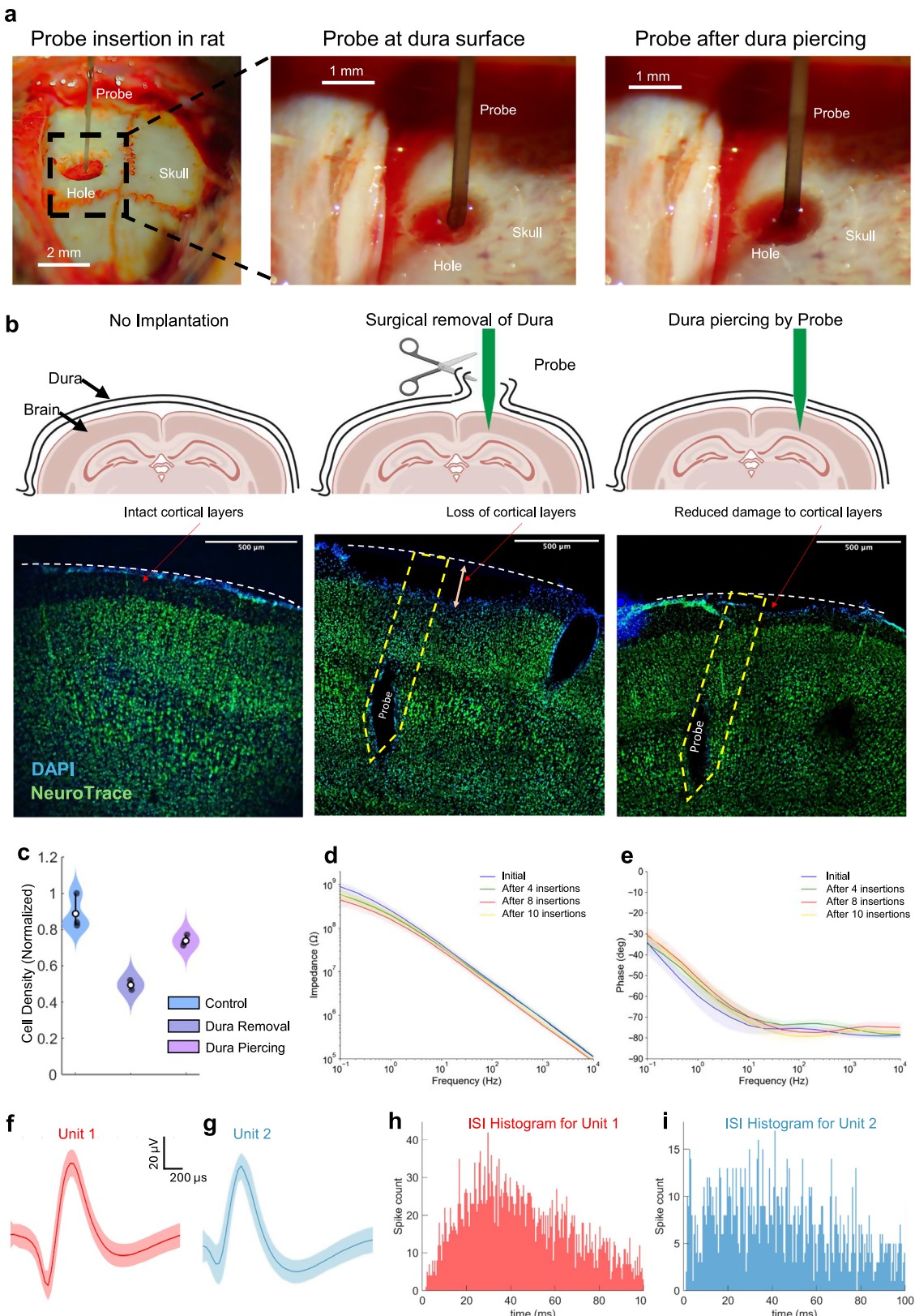

**c** Cell Density (Normalized): Control, Dura Removal, Dura Piercing

**d** Impedance (Ω) vs Frequency (Hz): Initial, After 4 insertions, After 8 insertions, After 10 insertions

**e** Phase (deg) vs Frequency (Hz): Initial, After 4 insertions, After 8 insertions, After 10 insertions

**f** Unit 1   **g** Unit 2   **h** ISI Histogram for Unit 1   **i** ISI Histogram for Unit 2

flexible probe architectures for non-human primates. This design is particularly promising for chronic applications, as the flexible tether cable accommodates brain micromotions and can significantly reduce tethering forces, potentially minimizing chronic tissue damage, necessary for demonstrating steeltrode's long-term recording performance. Rigid backends of steeltrodes allow simplified and high yield

packaging with commercially available ZIF connectors. The arrangement and pitch of the bondpad arrays can be easily changed in our design to accommodate other packaging schemes such as wirebonding, flip-chip bonding or direct interfacing with compact connectors like those from Omnetics[77], Molex[78] or Samtec[79], which are commonly employed in neural probe applications.

**Fig. 8 | Dura piercing and neural recording with steeltrode in rat brain.**
**a** Micrographs showing different stages of steeltrode implantation through rat dura. **b** Histological evaluation of intact cortical layers of brain without probe implantation, with probe implantation followed by surgical dura removal, and after dura puncturing with steeltrode. The coronal sections from brain were stained by nuclear stain- DAPI and fluorescent Nissl stain-Neurotrace to demonstrate neuronal density around the insertion site (created in BioRender. Chamanzar, M. (2025) https://BioRender.com/8bfbsk2). **c** shows comparison of normalized cell density across three conditions: Control (0.8875 ± 0.0564), Dura Removal (0.4944 ± 0.0264), and Dura Piercing (0.738 ± 0.0176). The values are represented as mean ± SEM (standard error of mean). Each violin represents the full distribution of data for individual hemispheres. Black dots indicate individual data points ($n = 3$

for Control and Dura Piercing and $n = 2$ for Dura Removal. The central open circle marks the mean value, and the vertical black line shows the full range (minimum to maximum). The shaded violin width reflects the kernel density estimate of data distribution. **d, e** Average electrochemical impedance and phase of the steeltrode remain stable over repeated dura penetrations, with measurements taken after 0, 4, 8, and 10 insertions. Data are presented as mean ± SD across 15 channels. **f, g** Representative averaged waveforms (with error bands indicating ±SD) of two putative units recorded from the rat hippocampus using a steeltrode that underwent 10 dura insertions. The successful recording of spiking activity after multiple insertions demonstrates the steeltrode's reusability. **h, i** Corresponding inter-spike interval (ISI) histograms for putative Units 1 and 2, respectively.

We validated our steeltrodes through acute recording of high signal-to-noise-ratio auditory-evoked local field potentials and single and multi-unit spiking activity from different depths of auditory cortex in macaque monkey, which is a deep region of the brain known to be difficult to access. Implantation was achieved simply by mounting the steeltrode on a standard microdrive, advancing it through a pre-positioned cannula and 3D-printed guide-hole grid. In contrast, NHP silicon probes with long shanks demand additional guide-tube assemblies and mounting hardware[80], shank-reinforcement methods[81], and artificial-dura systems[15] to avoid fracture, substantially enlarging the implantation chamber footprint and complicating the procedure.

Utilizing the same microfabrication and manufacturing process, we have also designed and realized stainless steel electrical recording probes with shorter shanks (<2 cm) for smaller animals, i.e., rodents. In addition to high-fidelity recording of neural signals from rodent brain, this robust steeltrode can be used for directly puncturing the rat dura, obviating the need for surgical removal of dura before implantation, thus preserving a larger part of the superficial layers of cortex. We showed that even after extensive reuse and multiple insertions, the steeltrodes preserved electrode integrity and continued to record high-quality neural signals, attesting to their robustness. However, a systematic evaluation to determine the threshold at which signal fidelity deteriorates under chronic use or repeated acute implantations should be explored in future studies.

Microfabrication on stainless steel provides tremendous versatility and robustness in terms of recording electrode arrangement and density. Compared to manually assembled commercially available stainless steel probes for non-human primate experiments, which are limited to a maximum of 64 channels with a minimum pitch of 50 μm, our microfabrication approach enables significantly higher channel densities with complete customizability. We have demonstrated electrode pitches as small as 25 μm, allowing for a much higher spatial resolution. Furthermore, our fabrication process supports feature sizes as small as 1 μm, which enables electrode counts of 100 channels per single metal layer. This number can be further scaled up using the multilayer microfabrication approach presented in this work, allowing for an even greater number of recording sites without significantly increasing the probe footprint. The ultra-high spatial resolution recording from deep brain regions facilitated by steeltrode (with electrode pitch down to 25 microns) holds significant potential for improving cortical circuit models compared to larger pitch probes, for example, as reported in by Chien et al.[82]. Moreover, the high degree of customizability in electrode arrangement on the steeltrode platform—enabled by precise photolithographic and microfabrication processes—allows for the design of probes tailored to neuroscience experiments that require targeted interrogation of specific and isolated brain regions. It is feasible to microfabricate a large number of customized steeltrodes in parallel for simultaneous volumetric neuronal recording from multiple areas of the brain. We can also readily adapt our fabrication process to realize probes with multiple shanks. We should however note that the backend headstage arrangement

should be optimized for implanting multiple closely-spaced neural probes.

The multilayer planar microfabrication technique on our hybrid polymer-stainless steel that we demonstrated in this paper, not only can increase the number of electrical recording channels, but also can be used to integrate microfabricated optical waveguide arrays to implement multimodal neural probes for simultaneous optogenetic stimulation and recording in non-human primates. In our previous work, we have demonstrated high-density low-loss Parylene C flexible optical waveguides which can be readily integrated with our hybrid stainless steel polymer platform[53,83]. Such devices can open new avenues for optogenetic research in primates with high-density optical stimulation, which is not currently feasible with current state-of-the-art and commercially available probes.

While we have not yet discussed electrical stimulation with steeltrodes in this study, their platinum electrodes—especially when coated with PEDOT:PSS—offer high charge-injection capacity and low impedance, supporting effective stimulation. Moreover, our fabrication process is compatible with sputtered iridium oxide films (SIROF) and other surface-modification strategies, paving the way for robust, integrated electrical stimulation capabilities in future.

Our steeltrode platform is built on austenitic stainless steel substrate, which offers few orders of magnitude higher resistance to fracturing compared to the widely used silicon material. Our microfabrication process provides unprecedented customizability in terms of patterning high-density microelectrodes with arbitrary arrangements, compared to the manual assembly processes used in commercially available stainless steel probes. The method that we discussed in this paper to realize steeltrodes for neural recording can be readily extended to realize other functional medical implants with micron-size features on stainless steel. It can also be extended to other material platforms such as tungsten and molybdenum, which are robust and biocompatible materials, similar to stainless steel.

## Methods
### Fabrication process of steeltrode
**PDMS–parylene C-based fabrication process.** Commercially available stainless steel substrate is first spin coated with ~7-μm-thick PDMS (Supplementary Fig. S2). To achieve this thickness, PDMS is diluted with hexane at a ratio of 1:3 and then spin coated on stainless steel at a spin speed of 4000 rpm for 180 s. Then the wafer is put in vacuum chamber at 120 °C temperature to cure and degas for 3 h. A thin layer (7 μm) layer of Parylene C is deposited on the wafer. Parylene C is deposited by using SCS Labcoter- 2 (dimer mass 8.3 g, furnace setpoint 690 °C, vaporizer setpoint 135 °C, vacuum setpoint 35 mTorr). Then, metal traces are patterned lithographically on Parylene C (AZ 5214E photoresist). Metal stack (5 nm Pt/ 120 nm Au/5 nm Pt) is deposited by electron-beam evaporator. Then, lift-off of metal traces is done at room temperature by soaking the sample in Acetone for 2 h. Afterwards, another thin-film (7 μm) of insulating Parylene C is deposited with the same process parameters as the first layer. Then, we deposit

100 nm of Chromium (Cr) using electron-beam evaporator and then photolithographically define (AZ 4110 photoresist and AZ 400K developer) the probe outlines and the electrode openings. Then Chromium is etched using Cr 1020 Etchant. Cr was used as a hardmask for subsequent parylene C etch step, as photoresist has poor selectivity to Parylene C. We etch Parylene C to expose electrodes, bond-pads, and the probe outlines by $O_2$ reactive ion etching (RIE) (14.0 sccm $O_2$, 50 mTorr, 50 W RF). After etching, the Cr hardmask is stripped using Cr 1020 Etchant. For etching the underlying PDMS layer to define the device outlines, another Cr hardmask is deposited and patterned using the same parameters as mentioned previously. PDMS is etched via reactive ion etching (RF = 120 W, Pressure = 35 mTorr) in $SF_6$ (22.5 sccm) and $O_2$ (9 sccm) chemistry. The final step of our fabrication process is to release each of the probes from the stainless steel wafer with laser micromachining process. With a high-powered fiber laser, stainless steel is etched away along the outlines of the probe. Lithographically defined alignment marks on wafer and CAD designs are used to accurately align and trace the borders of the probe with laser.

**SU-8-based steeltrode fabrication with single layer metal.** SU-8 2010 is spin coated on stainless steel wafer at 7000 rpm to produce 7-μm-thick bottom insulation layer. Next, the device outlines are defined in this layer via photolithography. Then we pattern the metal layer using AZ5214E photoresist. Then, we use electron beam evaporator to deposit 5 nm Cr/120 nm Au/5 nm Pt and subsequent lift-off in acetone to realize the metal traces and electrodes. Next, another 7-μm-thick SU-8 layer is spin coated and subsequently patterned to expose the electrode openings and device outline. SU-8 layers are baked at 180 °C for 10 min after photolithography. Finally, individual stainless steel probes are released from the wafer using a laser micromachining process.

**SU-8-based probe with multi-layer metal.** Before each metal layer, we have spin coated diluted SU-8 2010 (SU-8 2010 to SU-8 Thinner ratio of 1.8 by weight) at 4000 rpm to form a 2.5-μm-thick film. Probe outlines and electrode openings for the previous metal layers were photolithographically defined. Each SU-8 layer is baked at 180 °C for 10 min. Next, we pattern the metal layer using AZ5214E photoresist. Then, we use electron beam evaporator to deposit 5 nm Cr/120 nm Au/5 nm Pt and subsequent lift-off in acetone to realize the metal traces and electrodes. These steps are repeated for each metal layer. A 2.5-μm-thick SU-8 layer is deposited and patterned to form the top insulation layer exposing the recording electrodes and contacts. Finally, individual stainless steel probes are released from the wafer using a laser micromachining process.

**Electrochemical etching process to implement flexible tether cable on steeltrode.** We have utilized an electrochemical etching process with oxalic acid to selectively etch stainless steel underneath the polymer insulation layers. In this process, we mask the implantable rigid portion of the shank with Crystalbond and immerse the part of the probe that we want to etch in 10% oxalic acid solution at 35 °C which is being stirred at 300 rpm. Then we apply positive voltage to the stainless steel substrate of the probe with respect to a stainless steel counter electrode and impose 0.3 A current to the circuit with a Kiethley sourcemeter. This electrochemical etching process is continued until the stainless steel shank exposed to Oxalic acid is etched away completely. Then the Crystalbond covering the implantable shank of the probe is removed in Acetone. In this process, we can get probes where the implanted portion of shanks can freely move with brain micromotions with little or no tethering force and can retain electrical connection to the recording amplifier through a highly flexible PDMS–Parylene C cable.

**Electropolishing of steeltrode sidewalls.** To achieve a smooth surface finish on the stainless-steel sidewalls while maintaining polymer insulation integrity, we utilized an eco-friendly electropolishing process in a NaCl solution, following a procedure adapted from Han et al.[84]. The electrolyte consisted of 500 mL pure ethylene glycol solution with 1 M NaCl. The setup included a titanium mesh counter electrode, an Ag/AgCl reference electrode, and a custom 3D-printed jig to precisely hold the steeltrode, ensuring a uniform distance between the steeltrode and the mesh counter electrode. The electropolishing process was conducted at 50 °C, with an agitation speed of 300 rpm, while voltage was applied using an Autolab PGSTAT302N system. An electropolishing voltage in the range of 5.3- 5.6 V was used to achieve polishing uniformity and prevent excessive material removal. This method produced a smooth finish while maintaining the structural integrity and electrical functionality of the steeltrode.

**Electrochemical impedance characterization and electrode surface modifications.** Electrical performance of the probes was characterized by Electrochemical Impedance Spectroscopy (EIS) measurements in 1X PBS (Phosphate-Buffered Saline) solution. The characterization was conducted with a PGSTAT302N potentiostat (Metrohm, Netherlands) with a 3-electrode configuration in potentiostatic mode. The counter/auxiliary electrode was a Platinum wire electrode (MW- 1032, BASI Inc) and the reference electrode was Ag/AgCl electrode (MF- 2052, BASI Inc.). Electrochemical impedances of different channels of a typical probe were measured over the frequency range of 0.1 Hz–100 kHz. We used 50 mV (rms) sinusoidal signal at open circuit potential (OCP) to do the electrochemical measurements over this frequency range.

**PEDOT:PSS electrodeposition.** Conductive polymer PEDOT:PSS was electrochemically deposited on the surface of bare Pt electrode from an aqueous solution of 0.5 M 3,4-ethylenedioxythiophene (EDOT, Sigma-Aldrich, USA) and 0.6 wt% of poly(sodium 4-styrenesulfonate) (PSS, Sigma-Aldrich, USA). A 3-electrode configuration with Pt counter electrode and Ag/AgCl reference electrodes was used under galvanostatic condition. For each of the microelectrodes, conductive polymer is electrodeposited by applying constant current of 1 nA, until a charge density of 1 μC/cm² is reached. Each electrode was inspected optically and through EIS measurement to ensure successful electrodeposition.

**Probe bending experiment.** Housing for the backend of the probe was 3D printed using a polylactic acid (PLA) based filament. Probe was attached to the housing using UV curable resin (Piccasio UV Resin Clear) and placed inside a holder. Indenter was also 3D printed using PLA filament and taped to a motorized linear stage (Yun Duan model T06). Stage was driven by an open loop stepper motor (NEMA11 size) and controlled by a stepper motor controller (TB6600 based microstepping driver) and microcontroller development board (Arduino Nano). The linear stage was moved until the desired displacement of the probe tip was achieved. Later, the indenter was returned to initial position while the probe returned to its original shape.

## Animal experiments
### In-vivo measurements in macaque
**Animal model and surgical preparation.** All experiments were conducted in compliance with the Institutional Animal Care and Use Committee (IACUC) at the University of Pittsburgh. A single adult *Macaca mulatta* was used for acute intracranial electrophysiology, which remained awake throughout all the experiments discussed in the paper. To enable head fixation during recording sessions, a stainless-steel headpost (Christ Instruments) was surgically implanted, along with a cranial recording chamber over the frontal cortex. The recording chamber had been in place for over a year prior to the

steeltrode experiments to ensure long-term stability and accessibility to the cortical surface.

**Recording chamber grid and microdrive assembly.** A custom 3D-printed grid was designed to fit tightly into the recording chamber and provide precise alignment for guide tubes and electrodes. The grid was secured using set screws, ensuring stability throughout the recording session. A horizontal ledge was integrated into the grid to mount a microdrive, which was affixed with an additional set screw. The microdrive arm featured a micro-clamp to hold the steeltrode securely in place.

To maintain sterility, the microdrive assembly was first tested in a mock recording chamber before being sterilized using ethylene oxide. On the day of the experiment, the sterilized microdrive–grid assembly was transferred to the implanted recording chamber on the animal's head, ensuring aseptic conditions. A 23-gauge extra-thin-walled stainless-steel cannula (guide tube) was inserted through a selected guide hole to puncture the dura, after which the steeltrode was fed into the guide tube and aligned with the grid.

**Electrode insertion and recording procedure.** Once properly positioned, the steeltrode was secured to the microdrive, and its position was adjusted to align the electrode shaft with the grid hole. The steeltrode was then advanced in 0.5–2.0 mm increments using the microdrive until it reached the target depth within the auditory cortex. The entire recording procedure took place inside a sound-attenuating booth (Eckel) to minimize external noise interference. Neural signals were acquired using an Intan RHD 256-channel recording system (Intan Technologies, Los Angeles, CA, USA) at a sampling rate of 30 kS/s. The signals were referenced to the stainless-steel substrate of the probe. Initial recordings were performed to assess spontaneous neural activity, after which auditory stimuli were presented via free-field speakers to confirm probe placement within the auditory cortex. Local field potential (LFP) and multi-unit activity (MUA) data were analyzed using custom-developed MATLAB 2024a scripts. Single-unit data were processed and analyzed using custom Python scripts developed based on the Mountainsort[69].

**In-vivo measurements in rat.** All rodent experiments conducted in this study received ethical approval from the Institutional Animal Care & Use Committee (IACUC) of Carnegie Mellon University. All procedures were conducted as acute experiments, where probes were implanted and recordings performed during a single terminal session under anesthesia. For probe impanation, 3-month-old Sprague Dawley (SD) rats of both sex within a weight range of 200–300 g were used. To begin the experiments, rats were anesthetized with Ketamine/Xylazine cocktail (80 mg/kg and 8 mg/kg body weight) and placed on a feedback-controlled heating blanket maintained at 36 °C (Kent Scientific, Torrington, CT) mounted on a stereotaxic frame (Neurostar, Tubingen-Germany). An incision was made in the scalp, and craniotomy was performed by drilling a 1 mm diameter hole above one hemisphere with the coordinates AP −2.5 mm and ML + 3 mm targeting the somatosensory cortex and hippocampus. A screw hole was implanted on other hemisphere to provide a reference ground to the recording probe. Following craniotomy, a steeltrode probe was inserted at the target site at a depth of DV −3.5 mm to target CA1 regions of the hippocampus using a motorized stereotaxic arm at a speed of 100 μm/s. Following insertion, 10 min recovery was allowed for the acclimatization of tissue to the external material before starting spontaneous recordings. Spontaneous response was recorded for twenty minutes at 30 kHz sampling rate using a neural recording amplifier (Intan Technologies, Los Angeles, CA, USA). Spike band (single and multi-unit activity) data were processed and analyzed using Offline Sorter from Plexon Inc.

For histological evaluation, rats were transcardially perfused with fixative after 2 h of implantation. We transcardially perfused rats with PBS followed by 4% paraformaldehyde (PFA) to fix the tissues. Brain was isolated and kept in 4% PFA overnight followed by sequentially incubating in gradient sucrose solution (10%, 20% and 30%) for cryo-sectioning. Then 30 μm sectioned slices were prepared using cryostat and stored in PBS. Sections were further stained using NeuroTrace™ 500/525 Green Fluorescent Nissl Stain (Cat No. N21480, Thermofisher scientific), colabelled with DAPI (Nuclear Stain) and mounted on slides. To assess the effect of insertion on dura, images were captured at insertion site with the region of interest (ROI) measuring 900 μm × 500 μm covering dura and superficial cortical layers. Images were analyzed to measure cellular density using ImageJ and data was represented as normalized cellular density in each condition. Histology data was averaged for multiple insertions and normalized to max of control group.

**Probe cleaning.** After explanting the steeltrode at the end of in-vivo experiment, it is first rinsed off with Deionized (DI) water. Then the probe tip is immersed in 1% Tergazym solution for 2 h and then rinsed off again with DI water.

**Statistical power, replication, and exclusion criteria (rodent).** As no prior data were available to estimate sample sizes for this exploratory study, a formal a priori power analysis was not feasible. A total of four rats were used based on feasibility, ethical considerations, and alignment with similar proof-of-concept studies aimed at evaluating experimental parameters and generating preliminary data to inform future, adequately powered investigations.

Different hemispheres from each rat served as an independent biological replicate, with hemispheres allocated across three experimental conditions: three hemispheres for control (no insertion), two for dura removal, and three for dura piercing. This within-animal design maximized the amount of experimental data obtained while minimizing overall animal use. Only one insertion was performed per hemisphere to prevent localized tissue effects from multiple penetrations.

Two additional animals were excluded prior to histological analysis due to physiological instability during the procedure, including irregular respiration and inconsistent anesthesia depth despite adjustments. Data from the remaining four rats were included in all reported analyses.

Anesthesia depth and physiological parameters were carefully monitored to ensure procedural consistency, and all efforts were made to minimize both technical and biological variability. Quantitative histological comparisons were conducted at the level of individual tissue sections (6–9 per hemisphere), treated as technical replicates. Follow-up studies with larger sample sizes are planned to refine the quantification of these findings and explore condition-specific effects more comprehensively.

## Ethics Statement

The authors declare that the research was conducted in accordance with ethical guidelines and institutional regulations. All procedures involving animals were approved by the Institutional Animal Care and Use Committee (IACUC) at the respective research institutions and complied with relevant national and international ethical standards for the care and use of laboratory animals.

## Reporting summary

Further information on research design is available in the Nature Portfolio Reporting Summary linked to this article.

# Data availability

The raw datasets related to the key results presented in this manuscript are publicly available through the KiltHub repository at https://

doi.org/10.1184/R1/30505634.v1. The source data for generating the figures presented in this manuscript are provided with the paper. Any additional requests for information can be directed to, and will be fulfilled by, corresponding author. Source data are provided with this paper.

## Code availability

The custom analysis codes along with the relevant datasets used to generate the key results presented in this manuscript are publicly available through the KiltHub repository at https://doi.org/10.1184/R1/30505634.v1. The source data for generating the figures presented in this manuscript are provided with the paper.

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

## Acknowledgements

This work was supported by National Institutes of Health (NIH) grant 1RF1NS113303-01. Micron Laser Technology provided precision laser micromachining service. The authors acknowledge use of the Bertucci Nanotechnology Laboratory at Carnegie Mellon University supported by grant BNL-78657879. The authors also acknowledge and thank Jay Reddy, Xiang Li, Esther Bedoyan, and other contributors from the Chamanzar Research Laboratory for developing and maintaining the custom spike-sorting and neural signal-processing pipeline used to analyze the spike-band data from non-human primate and rodent experiments.

## Author contributions

Z.A. designed and fabricated devices; conducted benchtop characterizations and device packaging; designed in vivo experiments; analyzed non-human-primate and rodent in vivo data; performed mechanical simulations; drafted the manuscript and led the revisions. I.K. fabricated devices and performed packaging; conducted bending experiments and Electrochemical Impedance Spectroscopy characterizations. V.J. performed rodent in vivo experiments; conducted histology; analyzed histology data. K.G. performed non-human-primate experiments. T.T. designed and conducted non-human-primate experiments and associated stimulation/recording protocols; co-supervised the project. M.C. conceived the idea; designed experiments; supervised the whole project; and contributed to preparing the results and the manuscript. All authors discussed the results and approved the final manuscript.

## Competing interests

The authors declare no competing interests. M.C. and Z.A. are inventors on a pending U.S. patent application (US20220043028A1) related to this work. The remaining authors declare no competing interests.
