## [Transparent Peer Review file · Nature Communications]

Steelrode: Robust Minimally-invasive Microfabricated Stainless Steel Neural Interfaces for High Resolution Recording

Corresponding Author: Dr Maysamreza Chamanzar

Version 0:

Reviewer comments:

Reviewer #1

(Remarks to the Author)

Title: Steelrode: Robust Minimally-invasive Microfabricated Stainless Steel Neural Interfaces for High Resolution Recording.

This innovative work presents a new compact and high-density neural probes which can be implanted in different areas of the brain of rodents and non-human primates (NHP). While silicon has been the material of choice for rigid neural probes for rodents, its brittleness, and fragility are problematic for long aspect ratio silicon probes for large animals. Stainless steel is presented in this work as a biocompatible and less brittle material able to create robust neural probes, especially for large animal and human recordings. Moreover, stainless steel is more resilient and has a high modulus of elasticity. However, its microfabrication and micromachining technology has been less developed and is more challenging.

The authors discuss in this manuscript a novel microfabricated customizable stainless steel neural probe design for high-resolution recording in large animals. A rigid implantable stainless-steel shank and a flexible cable to relieve tethering force from the skull fixture have been created for this new type of probe. The design of high-density stainless steel and their implementation using a novel scalable microfabrication process can be repeated in multiple layers increasing the density and the number of recording electrodes.

These new probes have been tested using high-resolution in vivo neural recordings of single and multiple units across cortical laminae from the auditory cortex in rhesus macaques. Safe electrode implantation through intact dura in rats has been proven to record high action potentials, proving the robustness of electrodes for recording high-fidelity neural signals on rodent and NHP models to long-term chronic electrophysiology recording.

The work demonstrates the potential of stainless steel to be translated to humans and the possible applicability in epilepsy source localization and deep-brain-stimulation (DBS) electrode implantation.

The article is easy to read and well-explained. This new type of probe is innovative and it can be a great advance in animal research, especially with non-human primates. Researchers urgently need robust electrodes that simplify and make less time-consuming the experimental part and data acquisition. Thus, I find this work useful and interesting.

I have some comments and questions for the authors:

- Abstract: I think you should avoid citation sources in the abstract. But if you want to keep it, I think it should be cited in the Introduction part too.
- Page 4: "long enough to record from anywhere in the non-human primate brain". Later in the article has been written that the 8 mm electrode can raise the auditory cortex, as the experiment showed. It is not clear to me if this type of electrode allows recordings into the brainstem.
- Page 5: "The steelrodes were able to reliably record single unit and multi-unit activity as well as local field potentials ((Figure 1I)". There are two parenthesis before the word Figure.
"In contrast, since steelrodes are resilient against fracture, they can be safely inserted into the neural tissue multiple times and their handling and manipulation are reliable and safe. Moreover, these robust neural probes can be implanted through

the dura, obviating the need to remove dura, thus reducing damage to the superficial layers of the neural tissue". It is a good thing to have the possibility to insert the electrodes multiple times. However, if the penetrations are too close it can damage the tissue and create undesirable effects or symptoms in the animal. Reading all the article I understood that dura removing can be avoided only in rats, but not in NHP. However, this sentence is not specific.

- Page 8: "As a result, steeltrode itself can be used to puncture the dura of small animals during implantation, without the need for surgical removal of the dura. The ability to puncture through the intact dura layer without surgical removal is critical both for retaining the cranial pressure and for reducing the volume of removed tissue for probe implantation". I agree about the intracranial pressure undesirable effects when a dura removal is done. However, in my opinion, penetrating directly with the electrode can also damage the tissue with the pieces of dura that can go into the brain while the electrode is going in. This will contaminate the results and perhaps damage the electrode.

- Page 11: "For passive electrical neural probes, scaling up he the number of channels means increasing the number of traces connected to each recording electrode". Is the "he" an error in this sentence?
The citation 42 is cited two times in the same paragraph.

- Page 15: Things to keep in mind about the "brain micromotions". Sometimes the monkeys make pressure from the inside of the brain, making force with their bodies. Thus, they increase their intracranial pressure. When this happens the brain goes up and cells are killed and the brain tissue is detached from the probe influencing the data.

- Page 22: How many monkeys have been used to test the probes? Sometimes the animals have different behaviors that can influence the data or the probe's brittleness. For example, if it is a big male with aggressive behavior it will be more challenging to keep the electrode safe than if it is a female, even if both are head fixated.
How much time the researcher spent to insert the electrode into the brain?
What was the total duration of the experiment? (This has an influence on the behavior of the monkey as I explained above)?
How many times the probes can be used while keeping their data quality threshold?
How the electrode should be cleaned?
Did you check it into subcortical areas?

- Page 31: "Histological examination demonstrated that surgical removal of dura reduces the neural density along the tract...". When did this histological examination take place? The same day of the procedure or days later?

- Page 33: "Such devices can be implanted into deep cortical regions while being free to move with brain micromotions ensuring minimized chronic tissue damage due to significantly reduced tethering force". At the beginning of the article is written that electrodes can arise anywhere in the brain. It would be interesting to be precise if subcortical regions such as the brainstem can be recorded.

- Page 34: "We can also readily adapt our fabrication process to realize probes with multiple shanks". In this case, you should keep in mind the number of probes that the researcher uses at the same time. The arrangement of the instrument around the recorded zone has to have enough place and to be separated enough to allow several probes at the same time. Moreover, the position degree will influence if the shanks will touch each other.

Reviewer #2

(Remarks to the Author)

Key results:

This paper demonstrates the novel fabrication process to integrate PDMS / Parylene C / SU8 and platinum electrodes (that can be vertically stacked to increase channel count) on commercially available stainless steel. This process allows fabrication of high aspect ratio neural implants that are less fragile for implantation in vivo. The authors showcase the recording ability of such an implant in the macaque brain for auditory cortex recordings.

Main comment:

The claims that high-aspect ratio, high-density, versatile, stainless-steel probes, fabricated using microfabrication/micromachining processes, that can record with high-resolution from deep regions of the brain is supported by the data presented in the paper. However, the robustness/resilience against fracture of the stainless-steel probes is not thoroughly demonstrated in the present version of the manuscript, in my opinion. A figure does show the force required to fracture/plastically deform silicon vs. stainless steel, but no/little demonstration of what happens to the integrity of the electrodes integrated on the stainless-steel under stress, before/after bending, after multiple insertions, etc., is shown. "Reusability" of such devices, or "safety for multiple insertions" is not clearly demonstrated here.

Data and methodology questions/comments:

Main, P.5: "In contrast, since steeltrodes are resilient against fracture, they can be safely inserted into the neural tissue multiple times and their handling and manipulation is reliable and safe". It would be important to provide data to demonstrate this: the paper as it is does not really show "safe multiple insertion", nor that the probe can be safely manipulated. This would

be important, not only regarding the stainless-steel part, but also with the integrated electrodes, as part they are the critical part of the whole recording probe.

Main, P.7: "...whereas the steelrode only deforms and bends, thus causing much less and often reversible damage to the neural tissue": same here, what about the impact on the electrodes contact integrity of bending/deformation ?

Main, P.9: "We have fabricated probes with electrodes as small as 10- μm diameter and as large as 40- μm diameters.": I believe no data shows this directly in the manuscript. It could be interesting to see some sort of quantification, for example fabrication yield per electrode diameter (to give a sense of how many were successfully fabricated). "We have demonstrated a steelrode design with an inter-electrode pitch of as small as 25 μm for 10- μm diameter electrodes for high spatial resolution single unit recording. On the other end of the spectrum, we have also manufactured probes with sparsely distributed electrodes with larger diameters (30 – 40 μm), with pitches ranging from 1.5 – 2.5 mm. ": Similarly here, it would be interesting to see some supporting data to understand the fabrication challenges, and understanding fabrication outcomes for different types of constraints (small diameters vs. small pitches), if you have.

Main, P.11: "Such high-density metallic features can be used to realize more than 100 channels on a 250- μm wide stainless steel probe." Have you done any cross-talk measurements on this configuration?

Main, P. 22: "As a result, we have obtained ~40-fold reduction in impedance". Could you mention the actual impedance values in the result? Also, in Figure S3A, could it be possible to see the range from multiple contacts ? Is this just $n = 1$? Figure 4B: Would be interesting to see the impedance profiles for these recordings electrodes to see consistency of the spectrum over multiple contacts.

Results, P.31: "For assessing cellular damage, brain sections were stained using fluorescent Nissl stain (NeuroTrace) to evaluate neural density along the insertion tract. Histological examination demonstrated that surgical removal of dura reduces the neural density along the tract by 32% when compared to the group with no insertion, whereas in the case of probe insertion by piercing through dura, the reduction was only 18% when compared to control group (Figure 7C), thereby showing a significant improvement." I believe more details are needed here to evaluate the strength of these results. In particular: how many animals does this represent? And/or how many insertions? Error bar in Figure 7C suggests there was more than 1. When was the brain collected for analysis following insertion: right after, or after some time (tissue settling)? How long after implantation were the probes removed: right away or after some time? Was the whole tract considered for calculation of neural density, or just the top around dura as shown in the images on Figure 7? If only the top, how did you exactly define the region of the "superficial cortical layers" ? Was the intensity (Figure 7C) normalized? If yes, how? Here also, it could be useful to have electrode integrity before and after insertion (or even multiple insertions through the dura), or mimicking insertion forces, buckling, bending, to show that indeed that they are robust to manipulation and insertion, as the claim is that "Even for rodents, it is highly desired to design robust neural probes that can be safely used for multiple insertions."

Clarity and context questions/comments:

Structure suggestion: Nature Communications probably allows for quite flexible paper structures, but I would suggest some text restructuring to make the manuscript clearer for the reader to navigate and find useful information when needed:

- keep the "Main" free of specific results (the different design constraints are interesting to mention), but actual results of what was achieved can be kept for the result sections

- keep the result section free of methods (except the main elements for fabricating the steelrode, which are part of the main results of this paper, but for the electrophysiology and in vivo, I would suggest focusing on the actual results of the recordings and keep the methodology for the method section). This would alleviate the text and help have a more direct look at what was achieved, and the specifics of how it was done in the methods.

Abstract, P.1, last line: "from auditory cortex in rhesus macaques." : was this done in multiple animals? If not, keep "macaque" singular, to be precise.

Abstract, P.2, "to long-term chronic electrophysiology recording": no demonstration of long-term implantation of these probes is demonstrated in this paper, from what I could understand, so I would refrain from saying that these experiments "showcase the robustness of our steelrodes [...] in a gamut of applications ranging from [...] to long-term chronic". Or mention that this is the first step towards future experiments for long-term chronic electrophysiology for next generation BMI.

Main, P.2: "NHP probes have extremely high aspect ratios and thus need to be very rigid": they have to be rigid in the case where no guide/shuttle is used, but they don't need to be per se. I would maybe rephrase or add the mention that this applied for probes implanted without inserter/shuttles, to be more specific.

Main, P.4: "The steelrodes discussed in this paper address this issue by enabling high resolution distributed neural recording from different areas of the brain as envisioned in Figure 1A, where multiple steelrodes are implanted at different depths within the brain. Each of the penetrating neural probes is a hybrid polymer-stainless steel neural probe with a rigid shank that can be designed to be fully stiff (Figure 1B) or monolithically connected to a flexible cable (Figure 1C and 1D) to route the recorded signals to the backend circuitry, while minimizing the tethering force and damage to the brain tissue due to brain micromotions": This is the vision here, to have multiple steelrodes implanted, but from what I understand, this hasn't been done directly in this paper. So I would not say that the steelrodes "enable high resolution distributed neural recording from different areas", but they do definitely lead the path towards this vision. Make clearer that this is the vision, i.e. goal, but not demonstrated specifically in this paper. This paper (and if not, perhaps make it clearer) shows single insertion of a single probe with a fully rigid shank (Fig. 3A).

Main, P.5 : ((Figure 1E – 1G)) and ((Figure 1I) (extra parentheses).

Main, P.5: "As a demonstration of the capabilities of the platform, steelrodes with long implantation length (8 cm) with a very thin cross section of 140 μm x 280 μm with high density microelectrodes": Could you add the actual number of electrodes + pitch also here?

Main, P.6: "With these dimensions, the aspect ratio of our longest steelrode for NHPs (defined as the ratio of shank length to the thickness) would be ~570.": In order to better compare, can you give the equivalent aspect ratio of other relatable probes ?

No reference to figure 1H in the main text ? Also for Figure 1H, would be nice to have important dimensions directly highlighted, such as thickness, width, main materials, pitch, for a more comprehensive figure.

In general, I would try to be more specific when using the term "reliable": with respect to what ? Fabrication yield, recording ability, re-usability, something else ?

Main, P.4: "The manual manufacturing of such neural probes puts constraints on the density of microelectrode channels (e.g., currently, up to 64 channels with a minimum of 50 μm)": Can you maybe comment again in the discussion/conclusion, on how your device compares in terms of max channel count and pitch for these stainless steel probes?

Main, P.15: "Due to its small form-factor packaging and narrow cross section, multiple steeltrodes can potentially be implanted at different depths of the non-human primate brain for simultaneous high-density neural recording from different brain regions (Figure 1A): (if) this has not been actually tested, I would suggest to move this to the discussion as possible future next steps, not in the result section.

Figure 2A: scale bars on the images

Figure 2: "Metal" ? Be more specific. Which metal ? Also, in the text, and everywhere it applies, be specific with which metals.

Results, P.22: Some details are missing, which are important to mention. One vs. multiple animal ? Single vs. multiple insertions ? Acute vs. chronic ? Awake vs. Sedated ? If sedated, give information on anesthetics. All of these should be at least accessible in the methods.

Figure 3 and corresponding methods: Can you provide more info on this microdrive and implantation grid ? How was it implanted ? When in time, compared to the insertion of the probe ? What is placed acutely, or kept chronically to allow insertions at multiple times ? Can you comment on how this microdrive + guide tube + implantation chamber compares with "the use of specialized equipment and hardware [...] for ensuring proper alignment of such probes" (ex. Neuropixel NHP) as mentioned in the introduction ?

Results, P.24: "We recorded spontaneous neural activity at multiple depths corresponding to different layers of auditory cortex using a 16 channel steeltrode with high density microelectrodes with 25- μm inter- electrode separation. Why only 16 channels ? Are these just the channels at the tip ?

Figure 3B: scale bar for dimensions.

Figure 4C: Colorbar scale missing.

Figure 4B,D : Y-scale missing.

Results, P.26: Tone duration and inter-stimulus duration for the TRF experiments ? This should be in the method section, actually.

Figure 5: Can the chosen channels used to show mapping (Frequency vs. Intensity) be somehow highlighted on the schematic of the probe? Also, did you get different best frequency for two electrode contacts when the probe was in the same position ? Showing this (or better highlighting in the current figure) could help understand how useful the high resolution (small electrode pitch) is in differentiating best frequencies between close-by contacts.

Figure 5 legend: Labels in legend C,D,E,F don't match figure.

Figure 5 E-F: Can the depth at which these were recorded be given ? Or at least the distance in the tissue between the two measurements ?

Figure 5: "normalized MUA response": be more specific in the methods how this was computed and especially normalized.

Figure 5D: Y-label "Response" ? Is the also the normalized MUA response?

Discussion, P.33: "as well as long-term chronic brain interfacing": there is no demonstration of chronic use of these probes here, unless I missed it. I would discuss this as a future next step, but not as something readily demonstrated here.

Discussion, P.33: "Using selective electrochemical etching of stainless steel, we have created completely flexible tether cables attached to rigid shuttles and demonstrated hybrid rigid-flexible probe architectures for non-human primates. Such devices can be implanted into deep cortical regions, while being free to move with brain micromotions ensuring minimized chronic tissue damage due to significantly reduced tethering force." This was fabricated, but not implanted and tested (or at least not shown in this manuscript). I would more clearly mention that this would be an interesting improvement to the architecture used in the present NHP recordings, especially for chronic implants, and that is technically feasible, and could/should readily be tested.

Discussion, P.33: "different depths of auditory cortex in macaque monkeys". If $n = 1$ monkey, keep "monkey" singular.

Figure S2: "metal": can you be more specific as to which ones ?

Figure S3: (c) legend should be labeled as (b).

What about stimulation using the steeltrode ? Could this be achieved ? Has it been tested ? It could be interesting to mention/discuss this in the discussion. To the authors' discretion.

References:

Main, P.2: missing some references to support the claim that very small cross-section is important to minimize tissue damage.

Main, P.2: Ref. 13 and 14 are actually the same. Also, more recent literature could be added and interesting to refer to: <https://doi.org/10.1016/j.jneumeth.2023.110016>

Figure 4: Could be interesting to put this into perspective/discuss with more recent literature using Plexon stainless steel electrode: <https://doi.org/10.1016/j.neuroimage.2023.120364>

My expertise:

Sections regarding the microfabrication and micromachining of the steeltrode themselves and how novel the approach is, are out of the scope of my expertise. I leave it to the other reviewers to judge the feasibility, soundness and novelty of this process.

Version 1:

Reviewer comments:

Reviewer #1

(Remarks to the Author)

I appreciate and thanks to authors about the responses to my comments. All of my concerns have been adequately addressed, and I have no further suggestions. I support the publication of the revised manuscript.

Reviewer #2

(Remarks to the Author)

Thank you for addressing my concerns. I believe the manuscript has gained in clarity and additional data that was previously missing to support some of the main claims.

I have only a few additional small comments/remarks:

P.6 last paragraph end: "while minimizing the tissue damage. Damage." : Typo, twice "damage".

P.7 I.3: "the buckling force for the will be 40mN": word missing after "the".

P.9 I.21: "For steeltrodes, the devised microfabrication ...": "devised"?

P.22: "Error reference not found" x2

Legend Fig. 8 C) / P.34 I. new paragraph / "Statistical Power, Replication, and Exclusion Criteria" section: Can you make more clear in all these sections what N you are mentioning (and make sure you mention the appropriate N): is it N = number of animals or N = number of insertions or N = number of cross sections, and how many insertions/animals/sections per group/condition (control, dura removal, dura piercing) for comparison analysis? You mention in your response "The histology data were collected from 5 different insertions in 3 animals. Two insertions were performed in each animal.": was the second insertion performed on another hemisphere for example to avoid tissue effect due to the first insertion when looking at histology ? What happened to the 6th insertion (if 2 insertions/animals) ?

Reference 30 and 56 are the same ?

Fig. 8 A: can a scale bar be added for the images ?

Reviewer #3

(Remarks to the Author)

The authors have provided a detailed and systematic procedure on fabrication of steeltrode used as a brain sensor. The design aspects of the probe seem fine in my view, though following minor issues may be considered.

- Though fabrication steps have been provided in details, still it is not very clear that how did the authors made the long shank which is inserted in the brain. some clear figure may be provided.

- Also, in the figure 2S the first and second figure are same. Please look into this.

Response to the reviewer's comments:

We thank the reviewers for the very helpful comments and constructive feedback. We performed new experiments, revised the manuscript and carefully addressed all the comments. These revisions have enhanced the manuscript and further substantiate the messages of this paper. Below, we summarize our revisions and responses to the specific points raised by the reviewers.

Reviewer 1:

Title: Steelrode: Robust Minimally-invasive Microfabricated Stainless Steel Neural Interfaces for High Resolution Recording.

This innovative work presents a new compact and high-density neural probes which can be implanted in different areas of the brain of rodents and non-human primates (NHP). While silicon has been the material of choice for rigid neural probes for rodents, its brittleness, and fragility are problematic for long aspect ratio silicon probes for large animals. Stainless steel is presented in this work as a biocompatible and less brittle material able to create robust neural probes, especially for large animal and human recordings. Moreover, stainless steel is more resilient and has a high modulus of elasticity. However, its microfabrication and micromachining technology has been less developed and is more challenging.

The authors discuss in this manuscript a novel microfabricated customizable stainless steel neural probe design for high-resolution recording in large animals. A rigid implantable stainless-steel shank and a flexible cable to relieve tethering force from the skull fixture have been created for this new type of probe. The design of high-density stainless steel and their implementation using a novel scalable microfabrication process can be repeated in multiple layers increasing the density and the number of recording electrodes.

These new probes have been tested using high-resolution in vivo neural recordings of single and multiple units across cortical laminae from the auditory cortex in rhesus macaques. Safe electrode implantation through intact dura in rats has been proven to record high action potentials, proving the robustness of electrodes for recording high-fidelity neural signals on rodent and NHP models to long-term chronic electrophysiology recording.

The work demonstrates the potential of stainless steel to be translated to humans and the possible applicability in epilepsy source localization and deep-brain-stimulation (DBS) electrode implantation.

The article is easy to read and well-explained. This new type of probe is innovative and it can be a great advance in animal research, especially with non-human primates. Researchers urgently need robust electrodes that simplify and make less time-consuming the experimental part and data acquisition. Thus, I find this work useful and interesting.

I have some comments and questions for the authors:

- Abstract: I think you should avoid citation sources in the abstract. But if you want to keep it, I think it should be cited in the Introduction part too.

Response: Citations are removed from the abstract and we have added those citations to the introduction

- Page 4: “long enough to record from anywhere in the non-human primate brain”. Later in the article has been written that the 8 mm electrode can raise the auditory cortex, as the experiment showed. It is not clear to me if this type of electrode allows recordings into the brainstem.

Response: Depending on the approach angle, reaching the brainstem requires traversing on the order of 5 cm of brain tissue. Hence, it is in principle possible to reach the brain stem. There are two issues to note here. (1) Most NHP recording setups impose a rather large ‘dead’ space built in, meaning that a sizable portion of the shank can not be inserted into the brain and has to stay out due to the arrangement of the microdrive and the attachments. (2) While the current steelrodes were 8cm long, the proposed fabrication platform can be used to manufacture much longer probes. For example, rather than using a 4-inch wafer, a 6-inch substrate can be employed to fabricate probes longer than 12 cm. Therefore, by optimizing the approach angle, reducing dead space, and using the longer probes, it is indeed possible to reach the entire brain, including brain stem.

The scalability of our proposed neural probe fabrication platform to make steelrodes with >12 cm long shanks are noted in the *Discussion* section.

Revision: (In *Discussion* Section)

“The 8-cm steeltrode easily enables reaching subcortical regions in macaques. In our experiments, the approach angle, chamber design and microdrive arrangement dictated that between 2 and 4 cm of the shank length remained out of the brain tissue, thus leading to an effective electrode length of 4 to 6 cm. Given that the dorso-ventral extent of the monkey brain is below 5 cm, the current setup would provide access to most of the brain, including most subcortical targets. A redesign of the setup can further increase the effective length to around 7 cm, thus providing access to the entire monkey brain. For use in larger species, it will also be possible to use longer probes. The length of the neural probe was limited by the size of the substrate (i.e., a 4-inch-diameter wafer) that we made for photolithography. Since we made this wafer out of large stainless steel sheets, the size of the wafer can be easily made larger (i.e., 6 inches or even 8 inches) to microfabricate neural probes, with implantable shanks of up to 18 cm, if needed.”

- Page 5: “The steelrodes were able to reliably record single unit and multi-unit activity as well as local field potentials ((Figure 1I)”. There are two parenthesis before the word Figure.

Response: Thank you for pointing it out. This has been corrected in the revised manuscript

“In contrast, since steeltrodes are resilient against fracture, they can be safely inserted into the neural tissue multiple times and their handling and manipulation are reliable and safe. Moreover, these robust neural probes can be implanted through the dura, obviating the need to remove dura, thus reducing damage to the superficial layers of the neural tissue”. It is a good thing to have the possibility to insert the electrodes multiple times. However, if the penetrations are too close it can damage the tissue and create undesirable effects or symptoms in the animal. Reading all the article I understood that dura removing can be avoided only in rats, but not in NHP. However, this sentence is not specific.

Response: We have updated the text to make this claim specific for rodent dura.

Revision: Here is the updated text:

(Page 5): “Moreover, these robust neural probes can be implanted through **rodent** dura, obviating the need to remove dura, thus reducing damage to the superficial layers of the neural tissue. ”

- Page 8: “As a result, steeltrode itself can be used to puncture the dura of small animals during implantation, without the need for surgical removal of the dura. The ability to puncture through the intact dura layer without surgical removal is critical both for retaining the cranial pressure and for reducing the volume of removed tissue for probe implantation”. I agree about the intracranial pressure undesirable effects when a dura removal is done. However, in my opinion, penetrating directly with the electrode can also damage the tissue with the pieces of dura that can go into the brain while the electrode is going in. This will contaminate the results and perhaps damage the electrode.

Response: Thank you for raising this important concern. Steeltrode design features a customizable probe tip angle/sharpness so that it can slice through the dura cleanly, thereby minimizing the accumulation and transport of dural fragments.

To address the concern about compromising electrode integrity in the dura piercing process, we conducted additional experiments to evaluate the probe’s robustness during repeated insertions through the rat dura. In these tests, the same probe was inserted 10 consecutive times in rats. We monitored the electrode impedance after several insertions and observed no significant variation, which confirms that the probe’s structural and functional integrity remains intact. Moreover, even after the 10th insertion through dura, the probe successfully recorded putative single and multi-unit activities, demonstrating high-fidelity neural recordings. These results indicate that any potential contamination from small dural fragments does not compromise electrode performance, addressing the concern about tissue damage and electrode degradation.

Revision: We have included these findings in the revised manuscript in the Results section in the subsection titled “***Steeltrode with smaller shank is rigid enough to puncture rat dura and remains fully functional and reusable after repeated insertions***”

- Page 11: “For passive electrical neural probes, scaling up the number of channels means increasing the number of traces connected to each recording electrode”. Is the “he” an error in this sentence? The citation 42 is cited two times in the same paragraph.

Response: Corrections were made to the revised manuscript.

- Page 15: Things to keep in mind about the “brain micromotions”. Sometimes the monkeys make pressure from the inside of the brain, making force with their bodies. Thus, they increase their intracranial pressure. When this happens the brain goes up and cells are killed and the brain tissue is detached from the probe influencing the data.

Response: We acknowledge this is an important consideration, and we plan to further investigate these effects in future studies to ensure that the interface between the steeltrode and brain tissue remains reliable under all physiological conditions. The text has been revised to reflect that both head movements and brain micromotions in monkeys contribute to tethering force induced tissue damage and data quality degradation for neural probes.

Revision: On Page 15:

“As a result, once inserted into the brain, the implanted shank of the stainless steel probe can move independently of the backend, reducing tissue damage and preserving signal quality by mitigating tethering forces from skull attachment during brain micromotions and sudden bodily movements (Polanco et. al., Biosensors (2016) and Harris et. al., Nat. Neuroscience (2016).”

- Page 22: How many monkeys have been used to test the probes? Sometimes the animals have different behaviors that can influence the data or the probe's brittleness. For example, if it is a big male with aggressive behavior it will be more challenging to keep the electrode safe than if it is a female, even if both are head fixated.

Response: For the experiments reported in this paper, steeltrode was tested on a large male macaque, weighing above 15 kgs. With properly restrained and head-fixated monkey, we have not encountered any difficulty with the implantation of steeltrode, regardless of the individual monkey's behavior on the day of the experiment. It is our understanding that animal movement may play a role with extremely brittle silicon probes. For the steeltrodes, this is no concern at all. There is a small amount of risk when the electrodes get mounted to the microdrive (this happens without the animal present). At that stage, it is possible that the experimenter drops the assembly and probes bend. But once the electrodes have been attached to the microdrive, the process of mounting them to the recording chamber and inserting them into the brain is without risk to the steeltrode.

How much time the researcher spent to insert the electrode into the brain?

Response: The insertion of the steeltrode into the macaque brain took approximately 15 minutes. This duration also includes the time required to connect the neural amplifier, ground wires, and other necessary components.

What was the total duration of the experiment? (This has an influence on the behavior of the monkey as I explained above)?

Response: The typical macaque experiments had an approximate duration of 7 hours.

How the electrode should be cleaned?

Response: After explanting the probe, it is first rinsed off with Deionized (DI) water. Then the probe tip is immersed in Tergazyme solution for 2 hours, and then rinsed off again with DI water. The cleaning procedure is added to the methods section related to the animal experiments

Revision: We have added the following information to the *Methods* section:

Probe Cleaning

After explanting the steeltrode at the end of invivo experiment, it is first rinsed off with Deionized (DI) water. Then the probe tip is immersed in Tergazym solution for 2 hours and then rinsed off again with DI water.

Did you check it into subcortical areas?

Response: Thank you for your question. As noted in our previous response regarding the feasibility of reaching the brain stem with steeltrodes, by optimizing our approach—specifically by reducing dead space in implantation hardware and employing extended-length steeltrodes—it is indeed very feasible to reliably target subcortical areas, including the brain stem. However, in this paper, we focus on demonstrating the feasibility and robustness of steeltrodes for recording from the entire cortical laminae.

How many times the probes can be used while keeping their data quality threshold?

Response: This is a very important point. We have observed that even after insertion of and re-use of the same probe multiple times in NHP brain, the electrode impedances and the recorded signal quality (i.e., SNR) did not significantly change. We note that after each insertion the electrodes were cleaned using DI water and Tergazyme and sterilized using ethylene oxide before each re-use. Despite these aggressive cleaning and sterilization processes, the performance and signal quality did not degrade.

We have also conducted additional experiments to evaluate the probe's robustness and performance under repeated use. These experiments involved 10 consecutive insertions through rat dura using the same probe. We observed no significant changes in electrode impedance after multiple insertions, validating both the probe's structural integrity and the microelectrodes' functional integrity during repeated use.

Given the stable electrode impedance, it is reasonable to infer that the neural recording noise floor and subsequent data quality would exhibit similar stability, considering the inherent correlation between noise floor and electrochemical impedance. Furthermore, the successful acquisition of putative single- and multi-unit activities following the 10th insertion indicates that the electrodes retained their capacity for high-fidelity neural recordings even after substantial reuse.

Revision: The results of these experiments are added to the manuscript under the section titled “***Steelrode with smaller shank is rigid enough to puncture rat dura and remains fully functional and reusable after repeated insertions***”

Apart from the aforementioned section, we have added the following text to the *Discussion* section:

“In this paper, we report that even after extensive reuse and multiple insertions, the steelrodes preserved electrode integrity and continued to record high-quality neural signals, attesting to their robustness. However, a systematic evaluation to determine the threshold at which signal fidelity deteriorates under chronic use or repeated acute implantations lies beyond the scope of this study.”

Page 31: “Histological examination demonstrated that surgical removal of dura reduces the neural density along the tract...”. When did this histological examination take place? The same day of the procedure or days later?

Response: Brain sample for histological assessment was collected 2 hrs following implantation of probe to study the acute tissue damage/response following insertions.

Revision: The following is added to the *Methods* section to clarify this point:

(Page 46) “For histological evaluation, rats were transcardially perfused with fixative after 2 hours of implantation. Briefly, rats were transcardially perfused with PBS followed by 4% paraformaldehyde (PFA) to fix the tissues. Brain was isolated following fixation and kept in 4% PFA overnight followed by gradient sucrose solution (10%, 20% and 30%) as preprocessing for cryosectioning. Further 30m sectioned slice was prepared using cryostat and stored in PBS. Sections were further stained using NeuroTrace™ 500/525 Green Fluorescent Nissl Stain (Cat No. N21480, Thermofisher scientific), co labelled with DAPI (Nuclear Stain) and mounted on slides.”

Page 33: “Such devices can be implanted into deep cortical regions while being free to move with brain micromotions ensuring minimized chronic tissue damage due to significantly reduced tethering force”. At the beginning of the article is written that electrodes can arise anywhere in the brain. It would be interesting to be precise if subcortical regions such as the brainstem can be recorded.

Response: The 8 cm steelrode we have demonstrated in this work could in principle reach the brainstem, since reaching the brainstem only requires traversing about 5 cm of brain tissue, depending on the approach angle. However, as mentioned in a previous response, there are two things to consider: (1) Most NHP recording setups impose a ‘dead’ length, meaning a large part of the shank can't be inserted into the brain, due to hardware constraints of the implantation rig. (2) The current 8 cm steelrodes can be made much longer with the proposed fabrication platform. By optimizing the approach angle, reducing dead space, and using longer probes, it's possible to reach the entire brain, including the brain stem.

Revision: To clarify this issue in the manuscript, we have added the following to the *Discussion* section of the manuscript:

“The 8-cm steelrode easily enables reaching subcortical regions in macaques. In our experiments, the approach angle, chamber design and microdrive arrangement dictated that between 2 and 4 cm of the shank length remained out of the brain tissue, thus leading to an effective electrode length of 4 to 6 cm. Given that the dorso-ventral extent of the monkey brain is below 5 cm, the current setup would provide access to most of the brain, including most subcortical targets. A redesign of the setup can further increase the effective length to around 7 cm, thus providing access to the entire monkey brain. For use in larger species, it will also be possible to use longer probes. The length of the neural probe was limited by the size of the substrate (i.e., a 4-inch-diameter wafer) that we made for photolithography. Since we made this wafer out of large stainless steel sheets, the size of the wafer can be easily made larger (i.e., 6 inches or even 8 inches) to microfabricate neural probes, with implantable shanks of up to 18 cm, if needed.”

Page 34: “We can also readily adapt our fabrication process to realize probes with multiple shanks”. In this case, you should keep in mind the number of probes that the researcher uses at the same time. The arrangement of the instrument around the recorded zone has to have enough place and to be separated enough to allow several probes at the same time. Moreover, the position degree will influence if the shanks will touch each other.

Response: We acknowledge the importance of probe spacing and the backend headstage arrangement when implanting multiple probe/shanks simultaneously. We will carefully consider these factors in our future investigations to ensure optimal multi-shank probe configurations.

Revision: We have revised the manuscript to mention this important point:

Page 39: “We can also readily adapt our fabrication process to realize probes with multiple shanks. We should however note that the while implanting multiple multi-shank probes next to each other for high-density neural recording, the backend headstage arrangement should be optimized”.

Reviewer 2:

Key results:

This paper demonstrates the novel fabrication process to integrate PDMS / Parylene C / SU8 and platinum electrodes (that can be vertically stacked to increase channel count) on commercially available stainless steel. This process allows fabrication of high aspect ratio neural implants that are less fragile for implantation in vivo. The authors showcase the recording ability of such an implant in the macaque brain for auditory cortex recordings.

Main comment:

The claims that high-aspect ratio, high-density, versatile, stainless-steel probes, fabricated using microfabrication/micromachining processes, that can record with high-resolution from deep regions of the brain is supported by the data presented in the paper. However, the robustness/resilience against fracture of the stainless-steel probes is not thoroughly demonstrated in the present version of the manuscript, in my opinion. A figure does show the force required to fracture/plastically deform silicon vs. stainless steel, but no/little demonstration of what happens to the integrity of the electrodes integrated on the stainless-steel under stress, before/after bending, after multiple insertions, etc., is shown. “Reusability” of such devices, or “safety for multiple insertions” is not clearly demonstrated here.

Response: We appreciate your detailed feedback regarding the need to further demonstrate the robustness and resilience of our stainless-steel steeltrodes. In the revised manuscript, we have conducted additional experiments aimed specifically at addressing your concerns:

Mechanical Stress Testing:

To demonstrate functional robustness, we performed bending experiments and measured electrode impedances before and after aggressive bending. The impedance values remained unchanged, indicating that the probe and microelectrodes did not degrade under stress. The details of these experiments are included in the manuscript in the section titled “***Bending Experiments Validate the Mechanical Durability and Electrical Performance of Steeltrode***”.

Reusability and in vivo Multiple-Insertion Trials:

To further demonstrate and validate the reusability of the probes, we conducted in vivo experiments in which the same steeltrode was used for ten consecutive dura insertions in rodents. The electrochemical impedance of the electrodes remained stable across insertions. The probe continued to record putative single unit activities, confirming no significant changes in impedance

or electrode failure and that both the substrate and the microelectrode array remain fully operational during multiple uses. The methodology and results of these experiments are now discussed in Section “**Steelrode with smaller shank is rigid enough to puncture rat dura and remains fully functional and reusable after repeated insertions**” in the revised manuscript.

Data and methodology questions/comments:

Main, P.5: “In contrast, since steeltrodes are resilient against fracture, they can be safely inserted into the neural tissue multiple times and their handling and manipulation is reliable and safe”. It would be important to provide data to demonstrate this: the paper as it is does not really show “safe multiple insertion”, nor that the probe can be safely manipulated. This would be important, not only regarding the stainless-steel part, but also with the integrated electrodes, as part they are the critical part of the whole recording probe.

Response: We have added a section on our new in vivo experiments (Section titled “**Steelrode with smaller shank is rigid enough to puncture rat dura and remains fully functional and reusable after repeated insertions**”), specifically demonstrating the ability to insert and reuse our steeltrodes multiple times without compromising probe integrity even after rather aggressive cleaning and handling steps. We performed 10 successive dura penetrations in four animals using the same 15 mm steelrode and measured electrode impedances before implantation and after 4, 8, and 10 insertions. We observed negligible impedance changes, indicating that integrated microelectrodes were functionally and mechanically intact and robust. Crucially, the probe maintained high-fidelity spiking recordings from the rat hippocampus, confirming not only structural stability but also sustained electrophysiological performance. These results directly support our claim that the steelrode can be repeatedly inserted, safely handled and manipulated, and remain highly reliable for neural recordings.

Main, P.7: "...whereas the steeltrode only deforms and bends, thus causing much less and often reversible damage to the neural tissue": same here, what about the impact on the electrodes contact integrity of bending/deformation ?

Response: In the revised manuscript (see "***Bending Experiments Validate the Mechanical Durability and Electrical Performance of Steeltrode***"), we have included a new set of experiments specifically designed to address this issue.

The following figure shows the experimental setup and the measurements.

Figure 3 Mechanical bending tests of the steeltrode.

(A) Photographs and schematic diagram of the experimental setup, showing the steeltrode clamped in a 3D-printed mount, where a motorized indenter is placed next to the probe tip. (B) Steeltrode is bent at the tip by 10, 20, and 40 mm using the indenter. (C) Electrochemical impedance magnitude (Average and standard deviation, $N = 25$) plots after 0, 10, 20, and 40 mm of deflection, revealing minimal changes across frequencies from 0.1 to 10⁴ Hz. (D) Impedance phase angle plots under the same deflection conditions, likewise showing consistent behaviour relative to the undeformed state. These results confirm that the steeltrode remains electrically stable despite substantial bending.

In this experiment, a 78 mm-long steeltrode was secured at its backend using a custom 3D-printed mount, while a linear actuator, controlled by a microcontroller, applied bending forces to the free tip of the probe. The tip was incrementally displaced by 10, 20, 30, and 40 mm from its neutral (undeflected) position, and the probe was then returned to neutral after each step. Following every bending cycle, the electrochemical impedances of all channels of the steeltrode were measured.

Our data (Figures 3(C) and 3(D) in the revised manuscript) shows that neither the impedance magnitude (1.55 ± 0.13 MΩ at 1 kHz) nor the phase ($-70 \pm 1^\circ$ at 1 kHz) deviated significantly from the baseline, even after the maximum deflection of 40 mm. This high level of stability indicates

that the microelectrodes—along with their insulation layers—remained structurally and electrically intact, with no detectable microcracking, delamination, or adhesion failures.

Main, P.9: “We have fabricated probes with electrodes as small as 10- μm diameter and as large as 40- μm diameters.”: I believe no data shows this directly in the manuscript. It could be interesting to see some sort of quantification, for example fabrication yield per electrode diameter (to give a sense of how many were successfully fabricated). “We have demonstrated a steelrode design with an inter-electrode pitch of as small as 25 μm for 10- μm diameter electrodes for high spatial resolution single unit recording. On the other end of the spectrum, we have also manufactured probes with sparsely distributed electrodes with larger diameters (30 – 40 μm), with pitches ranging from 1.5 – 2.5 mm. “: Similarly here, it would be interesting to see some supporting data to understand the fabrication challenges, and understanding fabrication outcomes for different types of constraints (small diameters vs. small pitches), if you have.

Response: Our fabrication yield is primarily influenced by the smallest feature size, which is the metallic trace width that determines the number of channels, rather than the electrode diameter itself.

In all of our designs, the trace width is chosen to be larger than $\sim 2 \mu\text{m}$. While this is a rather small trace width, it is a conservative choice for our microfabrication process. With this conservative choice, we can fabricate almost perfect neural probes for many practical applications in neuroscience. As shown in our paper, our lithography and lift off process can easily enable defining 1 μm traces (Figure 2) with an almost perfect yield. Smaller trace widths can lead to increased sensitivity to process variations.

Our optimized fabrication process has consistently produced high yields across the range of electrode diameters (10–40 μm). At this stage, given the almost perfect yield of our lithography and lift off process at the conservative choices of trace widths, we do not have enough datapoints to judge the yield in a statistical sense. In the future, we plan on pushing the limits on the trace width and also multi-layer processing (as discussed in Figure 2) to dramatically scale up the number of channels, in which case, we will conduct rigorous yield analyses for the lithography and also the lift off processes, separately and combined.

Revision: The following is added to the manuscript to emphasize the need for optimizing the yield when using very small feature sizes.

(Page 19): “When using narrower interconnects, especially in multi-layer designs, the fabrication process has to be optimized to achieve a high yield.”

Main, P.11: “Such high-density metallic features can be used to realize more than 100 channels on a 250- μm wide stainless steel probe.” Have you done any cross-talk measurements on this configuration?

Response: The theory and literature (including our previous work) suggest that crosstalk is negligible under these conditions. Crosstalk in planar microelectrode arrays is generally dominated by capacitive coupling between adjacent traces, which can be mitigated by appropriate

insulation thickness and separation distance. Our previous work has demonstrated that even sub-micron (≈ 250 nm) inter-trace gaps are feasible without appreciable crosstalk, particularly when using biocompatible polymer dielectrics such as SU-8 or Parylene-C with thicknesses on the order of several micrometers (see for example Qiang et al., Nano Research 14, 3240–3247 (2021), and Chamanzar et al., Proc. of IEEE MEMS (2015)). Given our relatively conservative $1\ \mu\text{m}$ spacing, relatively thicker insulation layers ($\geq 2.5\ \mu\text{m}$), the expected capacitive coupling between channels will be negligible.

Revision: The following is added to the manuscript:

(Page: 18): “With a $1\ \mu\text{m}$ space and trace, the crosstalk between adjacent traces over the entire length of the probe is minimal, as shown in the previously reported crosstalk analyses [Qiang et al., Nano Research 14, 3240–3247 (2021)].”

Main, P. 22: “As a result, we have obtained ~ 40 -fold reduction in impedance”. Could you mention the actual impedance values in the result? Also, in Figure S3A, could it be possible to see the range from multiple contacts ? Is this just $n = 1$?

Figure 4B: Would be interesting to see the impedance profiles for these recordings electrodes to see consistency of the spectrum over multiple contacts.

Response: We have updated the manuscript to include impedance value ($51.2 \pm 2.26\ \text{k}\Omega$ at 1kHz, for $N = 14$ channels) after PEDOT coating.

Figure S3 has been updated to include the mean and standard deviation (shaded region) of impedance magnitude and phase to indicate “mean \pm SD” for measurements from 14 channels ($N = 14$). The electrochemical impedance magnitude and phase across channels are very consistent, as can be seen from the very narrow standard deviation in Fig S3a (also shown below, for your convenience). Specifically, at a frequency of 1 kHz, the 15-micron diameter bare platinum (Pt) electrodes ($N = 14$) on a steelrode demonstrated a mean impedance magnitude of $1.07 \pm 0.093\ \text{M}\Omega$. In contrast, when the same electrodes were coated with PEDOT:PSS, the mean impedance magnitude decreased to $51.2 \pm 2.26\ \text{k}\Omega$, again across all the 14 channels. The Fig. S3 and the text are updated to clarify the details.

The updated Figure S3 is provided below:

Figure S3 Electrochemical impedance spectroscopy (EIS) measurements comparing bare Pt electrodes and PEDOT:PSS-coated electrodes. (a) Magnitude (mean \pm SD) and (b) phase (mean \pm SD) of impedance spectra ($N = 14$ channels). (c) Microscope image illustrating the coated and uncoated electrodes.

Revision: Manuscript is also revised to note the following:

(Page 22): “As a result, PEDOT:PSS coated electrodes ($N = 14$) on steelrodes exhibited impedances of 51.2 ± 2.26 kΩ at 1kHz (Figure S3 A), representing ~ 40 fold reduction compared to bare Pt electrodes.”

Results, P.31: “For assessing cellular damage, brain sections were stained using fluorescent Nissl stain (NeuroTrace) to evaluate neural density along the insertion tract. Histological examination demonstrated that surgical removal of dura reduces the neural density along the tract by 32% when compared to the group with no insertion, whereas in the case of probe insertion by piercing through dura, the reduction was only 18% when compared to control group (Figure 7C), thereby showing a significant improvement.” I believe more details are needed here to evaluate the strength of these results. In particular: how many animals does this represent? And/or how many insertions? Error bar in Figure 7C suggests there was more than 1. When was the brain collected for analysis following insertion: right after, or after some time (tissue settling)? How long after implantation where the probes removed: right away or after some time? Was the whole tract considered for calculation of neural density, or just the top around dura as shown in the images on Figure 7? If only the top, how did you exactly define the region of the “superficial cortical layers”? Was the intensity (Figure 7C) normalized? If yes, how? Here also, it could be useful to have electrode integrity before and after insertion (or *even multiple insertions through the dura*), or mimicking insertion forces, buckling, bending, to show that indeed that they are robust to manipulation and insertion, as the claim is that “Even for rodents, it is highly desired to design robust neural probes that can be safely used for multiple insertions.”

Response:

This section of the manuscript has been thoroughly revised to include the results from additional experiments that were conducted to address the reviewer’s concerns as well as detailed methodology is included for better understanding. These experiments:

- (1) strengthen our histological analysis, and
- (2) demonstrate the stability and integrity of the electrode after multiple dura insertions.

The histology data were collected from 5 different insertions in 3 animals. Two insertions were performed in each animal. Brain samples were collected 2 hr after the implantation was performed to study acute tissue damage/response. For analysis, only the superficial layers of cortex (with ROI $900 \mu\text{m} \times 500 \mu\text{m}$) were included to quantify damage in superficial layers due to dura removal (the areas of quantification kept constant in different sections). Data is represented as neuronal density/ mm^2 (mean \pm SEM) in selected ROI. The data is normalized to the control group.

We have performed new experiments to test the reusability of the steelrode by performing 10 insertions through the rat dura using the same probe. The electrode impedance was measured after every few insertions and it was observed that there was no significant variation in the impedance. Even after the 10th insertion, steelrode was able to record putative single and multi-unit activities. These results conclusively establish the robustness of the probe during multiple insertions.

Revision: Please refer to the section titled, “*Bending Experiments Validate the Mechanical Durability and Electrical Performance of Steelrode*” for details of these experiments.

Clarity and context questions/comments:

Structure suggestion: Nature Communications probably allows for quite flexible paper structures, but I would suggest some text restructuring to make the manuscript clearer for the reader to navigate and find useful information when needed:

- keep the “Main” free of specific results (the different design constraints are interesting to mention), but actual results of what was achieved can be kept for the result sections
- keep the result section free of methods (except the main elements for fabricating the steelrode, which are part of the main results of this paper, but for the electrophysiology and in vivo, I would suggest focusing on the actual results of the recordings and keep the methodology for the method section). This would alleviate the text and help have a more direct look at what was achieved, and the specifics of how it was done in the methods.

Response: We have restructured the manuscript to present the results more concisely and have moved the technical details related to methodology to the *Methods* section. The *Methods* section now includes detailed protocols and technical specifications pertaining to the animal experiments to enhance clarity and reproducibility.

Abstract, P.1, last line: “from auditory cortex in rhesus macaques.” : was this done in multiple animals? If not, keep “macaque” singular, to be precise.

Response: Corrections have been made to the manuscript.

Abstract, P.2, “to long-term chronic electrophysiology recording”: no demonstration of long-term implantation of these probes is demonstrated in this paper, from what I could understand, so I would refrain from saying that these experiments “showcase the robustness of our steelrodes [...] in a gamut of applications ranging from [...] to long-term chronic”. Or mention that this is the first step towards future experiments for long-term chronic electrophysiology for next generation BMI.

Response: We have removed the claim of "long-term chronic electrophysiology recording" from the abstract and have added text that mentions that this is a first step towards long term chronic electrophysiology recordings with next generation BMIs.

Revision: We have added the following revised text to the Abstract

“These in vivo experiments demonstrate the robustness of our steelrodes for recording high-fidelity neural signals in a range of applications, including basic science research on rodent and NHP models, and represent a crucial first step towards future long-term chronic electrophysiology studies for the development of effective therapeutics and next-generation brain-machine interfaces.”

Main, P.2: “NHP probes have extremely high aspect ratios and thus need to be very rigid”: they have to be rigid in the case where no guide/shuttle is used, but they don’t need to be per se. I would maybe rephrase or add the mention that this applied for probes implanted without inserter/shuttles, to be more specific.

Response: We agree with the reviewer that the rigidity requirement is specific to probes implanted without inserters/shuttles. We have modified the text to reflect this:

Revision: The following text is revised in the Main section:

“... a typical NHP probe requires substantial rigidity and robustness to endure the stresses of implantation and post-implantation when implanted without rigid inserters.”

Main, P.4: “The steelrodes discussed in this paper address this issue by enabling high resolution distributed neural recording from different areas of the brain as envisioned in Figure 1A, where multiple steelrodes are implanted at different depths within the brain. Each of the penetrating neural probes is a hybrid polymer-stainless steel neural probe with a rigid shank that can be designed to be fully stiff (Figure 1B) or monolithically connected to a flexible cable (Figure 1C and 1D) to route the recorded signals to the backend circuitry, while minimizing the tethering force and damage to the brain tissue due to brain micromotions”: This is the vision here, to have multiple steelrodes implanted, but from what I understand, this hasn’t been done directly in this paper. So I would not say that the steelrodes “enable high resolution distributed neural recording from different areas”, but they do definitely lead the path towards this vision. Make clearer that this is

the vision, i.e. goal, but not demonstrated specifically in this paper. This paper (and if not, perhaps make it clearer) shows single insertion of a single probe with a fully rigid shank (Fig. 3A).

Response: We appreciate the reviewer's feedback and have modified the text to accurately reflect the scope of the current work.

Revision: Here is the revised text:

(Page 5): "The steelrodes discussed in this paper provide a path towards high-resolution distributed neural recording from various brain areas, as shown in Figure 1A, where multiple steelrodes are implanted at different depths within the brain."

Main, P.5 : ((Figure 1E – 1G)) and ((Figure 1I) (extra parentheses).

Main, P.5: "As a demonstration of the capabilities of the platform, steelrodes with long implantation length (8 cm) with a very thin cross section of 140 μm x 280 μm with high density microelectrodes": Could you add the actual number of electrodes + pitch also here?

Response: The number and pitch ranges for electrodes in various steelrode designs are included.

Revision: Here is the revised text:

(Page 5): "As a demonstration of the capabilities of the platform, steelrodes with long implantation length (8 cm) with a very thin cross section of 140 μm x 280 μm with high density microelectrodes (16- 100 channels, with pitches ranging from 25 μm to 2.5 mm) were fabricated and tested in macaque auditory cortex which is located deep in the lateral fissure and is notoriously difficult to reach."

Main, P.6: "With these dimensions, the aspect ratio of our longest steelrode for NHPs (defined as the ratio of shank length to the thickness) would be ~570.": In order to better compare, can you give the equivalent aspect ratio of other relatable probes ?

Response: The aspect ratio of relatable NHP probes are provided in the text.

Revision: Here is the revised text:

(Page 6): With these dimensions, the aspect ratio of our longest steelrode for NHPs (defined as the ratio of shank length to the smallest cross-sectional dimension) would be ~570. This is an extremely high aspect ratio compared to existing probes for NHPs, for example Plexon V probe and Neuropixel (NHP-Long) probe have aspect ratios of ~307 and ~360, respectively. For an 8-cm long, 140 μm thick, and 260 μm wide steelrode, the buckling force for the will be 40 mN, whereas the buckling force for a silicon probe with the same dimensions would be 32 mN.

No reference to figure 1H in the main text ? Also for Figure 1H, would be nice to have important dimensions directly highlighted, such as thickness, width, main materials, pitch, for a more comprehensive figure.

Response: The caption for Figure 1H is updated to include the details of the dimensions of the probe and microelectrodes, and this figure is also referred to in the main text.

Revision: The following text is added to the manuscript:

(Page 15): "Figure 1H shows SEM image of tip of a steeltrode showing its smooth sidewalls, and 15- μm diameter microelectrodes with a pitch of 50 micron."

In general, I would try to be more specific when using the term "reliable": with respect to what ? Fabrication yield, recording ability, re-usability, something else ?

Response: We have carefully reviewed the instances where the term "reliable" is used and have provided more specific context in each case to clarify the intended meaning.

Revision: Few instances of the revisions are provided below:

"... In contrast, since steeltrodes are resilient against fracture, they can be safely inserted into the neural tissue multiple times and their handling and manipulation is ~~reliable~~ repeatable and safe."

"... Therefore, in addition to enabling ~~reliable~~—high-fidelity recording from deep brain regions of large animals, the novel microfabrication process on stainless steel discussed in this work can also enable mechanically robust stainless steel probe with smaller implantable shank for recording in small animals without the need for surgical removal of dura."

"... Therefore, stainless steel neural probes would be more robust and ~~reliable~~ resistant to fracturing. "

"... We can adapt our novel microfabrication process on stainless steel to implement **mechanically** robust and ~~reliable~~ steeltrodes with cm-long implantable shanks which are suitable for recording from rodents and other small animal brains."

"... This is a great advantage over the conventional methods of manufacturing the rigid probes and flexible cables separately and packaging them together, resulting in a more ~~reliable design~~—**streamlined** and a scalable fabrication process."

"... Such good water barrier properties are essential for **ensuring consistent signal quality** during ~~reliable~~—chronic electrical recording. ~~performance of neural interfaces.~~ "

Main, P.4: “The manual manufacturing of such neural probes puts constraints on the density of microelectrode channels (e.g., currently, up to 64 channels with a minimum of 50 μm) “: Can you maybe comment again in the discussion/conclusion, on how your device compares in terms of max channel count and pitch for these stainless steel probes?

Response: This point is reiterated in the *Discussion* section.

Revision: The following text is added in the *Discussion* section

“Compared to manually assembled commercially available stainless steel probes for non-human primate experiments, which are limited to a maximum of 64 channels with a minimum pitch of 50 μm , our microfabrication approach enables significantly higher channel densities with extreme customizability. We have demonstrated electrode pitches as low as 25 μm , allowing for a much higher spatial resolution. Furthermore, our fabrication process supports feature sizes as small as 1 μm , which facilitates electrode counts of 100 channels per single metal layer. This number can be further scaled up using the multilayer microfabrication approach presented in this work, allowing for an even greater number of recording sites without significantly increasing the probe footprint.”

Main, P.15: “Due to its small form-factor packaging and narrow cross section, multiple steelrodes can potentially be implanted at different depths of the non-human primate brain for simultaneous high-density neural recording from different brain regions (Figure 1A): (if this has not been actually tested, I would suggest to move this to the discussion as possible future next steps, not in the result section.

Response: Thanks for pointing it out. We have moved this sentence to the *Discussion* section.

Figure 2A: scale bars on the images

Response: Scale bars are added.

Figure 2: “Metal” ? Be more specific. Which metal ? Also, in the text, and everywhere it applies, be specific with which metals.

Response: In our manuscript, the term "metal" refers specifically to the metallic layers used for interconnects. These metallic traces consist of either a Cr/Au/Pt stack (for SU-8-based steelrodes) or a Pt/Au/Pt stack (for Parylene-C/PDMS-based steelrodes). Where appropriate, we have explicitly clarified the specific details and function of these metal layers within the manuscript.

Results, P.22: Some details are missing, which are important to mention. One vs. multiple animal ? Single vs. multiple insertions ? Acute vs. chronic ? Awake vs. Sedated ? If sedated, give information on anesthetics. All of these should be at least accessible in the methods.

Response: We are now adding a detailed *Methods* section that addresses these concerns and provides comprehensive information regarding the specifics of the animal experiments, including, but not limited to the number of animals, the number of insertions, and the anesthesia used.

Figure 3 and corresponding methods: Can you provide more info on this microdrive and implantation grid ? How was it implanted ? When in time, compared to the insertion of the probe ? What is placed acutely, or kept chronically to allow insertions at multiple times ? Can you comment on how this microdrive + guide tube + implantation chamber compares with “the use of specialized equipment and hardware [...] for ensuring proper alignment of such probes” (ex. Neuropixel NHP) as mentioned in the introduction ?

Response: The NHP recording setup used standard procedures for acute intracranial recordings. Briefly, a recording chamber was chronically implanted to provide access to the dura. The first recordings are typically made at least 2 weeks after the installation of the chamber. In this specific case, the chamber had been in place for well over a year prior to the steeltrode recordings. The recording setup consists of a 3D printed grid of guide-holes that tightly fits into the recording chamber and is fixed in place using several set screws. The 3D printed grid also contains a horizontal ledge onto which a microdrive is mounted with a set screw. The microdrive arm is equipped with a micro clamp to attach the electrode to the drive. The electrode is fed into a guide-tube which in turn is fed into the desired grid location. Next, the microdrive is attached to the electrode and its position on the ledge is adjusted to align the electrode shaft with the grid-hole. The microdrive is assembled in a mock recording chamber and sterilized using Ethylene Oxide. On the day of the recording, the assembly is transferred to the actual recording chamber on the animal's head. The guide-tube is pushed forward to penetrate the dura. Then the electrode shaft is advanced to the desired depth.

Implanting Neuropixels probes in non-human primates (NHPs) is challenging due to their fragile silicon shanks, which can break when penetrating the thick dura and overlying tissue growth. To prevent damage, guide tubes or cannulas are required, but maintaining precise alignment between the probe shank and guide tube is difficult, often necessitating specialized equipment. Minor misalignment can lead to probe fracture, making the process highly delicate. To address this, reinforcement techniques (Q. Wang et al, *NER* 2021) and artificial dura (Tomoyuki Namima et al, *Journal of Neuroscience Methods*, 2024) systems have been developed.

Another major limitation is the size constraint within the recording chamber. Neuropixels probes require a larger footprint than traditional microelectrodes due to their separate guide tube holding apparatus, which reduces the available targetable brain area and restricts flexibility in probe placement (Devyn Lee Bauer et. al, *JNE* 2023).

In contrast, steeltrode implantation is far simpler and more space-efficient. Steeltrodes are mechanically robust and can be directly guided through a pre-positioned cannula without the risk of bending or breaking, eliminating the need for complex alignment mechanisms. This makes the insertion process more straightforward, reliable, and efficient compared to Neuropixels probes.

Revision: The following text is added to the *Discussion* section:

“Implantation was achieved simply by mounting the steelrode on a standard microdrive, advancing it through a pre-positioned cannula and 3D-printed guide-hole grid. In contrast, NHP silicon probes with long shanks demand additional guide-tube assemblies and mounting hardware, shank-reinforcement methods, and artificial-dura systems to avoid fracture, substantially enlarging the implantation chamber footprint and complicating the procedure.”

Results, P.24: “We recorded spontaneous neural activity at multiple depths corresponding to different layers of auditory cortex using a 16 channel steelrode with high density microelectrodes with 25- μ m inter- electrode separation. Why only 16 channels ? Are these just the channels at the tip ?

Response: The recordings reported in this paper were carried out using high density 16 channel steelrodes due to the availability of the adaptor PCBs, headstages and the backend electronics to demonstrate the concept. All of the methods and procedures will directly translate to the higher-channel count versions that we have fabricated and tested using benchtop characterization methods.

Figure 3B: scale bar for dimensions.

Figure 4C: Colorbar scale missing.

Figure 4B,D : Y-scale missing.

Response: The manuscript and the corresponding figures have been updated to address all these comments.

Results, P.26: Tone duration and inter-stimulus duration for the TRF experiments ? This should be in the method section, actually.

Response: Tonal response fields (TRFs) were measured using a receptive mapping task. Animals passively listened to pure-tone pips of 50 ms duration (5 ms on and offset taper) presented at 32 different frequencies (equidistantly spaced in log₂ space between 90Hz to 25kHz) and presented at 5 intensities (20, 30, 40, 50, and 60 dB SPL). Time between tones varies between 0.8 and 1.2 seconds. Each combination of frequency and intensity was repeated 10 times. Information related to tone duration, and inter-stimulus interval is added to manuscript.

Revision: The following information with detailed stimulation parameters are added to the manuscript

(Page 28): “In our experiments using steelrodes, TRFs were measured with a standard receptive mapping task. Animal passively listened to 50-ms pure-tone pips with a 5-ms on/off taper (Figure 6A). Tones were presented at 32 frequencies that are equidistantly spaced in log₂ space between 90 Hz and 25 kHz, and at five intensity levels (20, 30, 40, 50, and 60 dB SPL). The interval between tones varied randomly between 0.8 and 1.2

seconds. Each frequency–intensity combination was repeated 10 times, resulting in an approximately 40-minute recording session.”

Figure 5: Can the chosen channels used to show mapping (Frequency vs. Intensity) be somehow highlighted on the schematic of the probe?

Response: The chosen channel is marked with an arrow in the schematic of the probe in the manuscript.

Revision: Following is the updated figure identifying the channel on the schematic of the probe for your convenience

Figure 6 Macaque tonal response field measurement with Steeltrode.

(A) Schematic of the stimulation paradigm used to generate the tonal response field. Tones with varying frequencies and amplitudes were presented at randomized intervals. (B) Trial-averaged laminar multi-unit activity (MUA) recorded from the macaque auditory cortex in response to a 7.1 kHz tone at 60 dB SPL. (C) Normalized MUA response calculated from the trial-averaged data for the channel showing the strongest response (indicated by the red arrow in B). This panel demonstrates frequency tuning at three different stimulus volume levels across frequencies from 90 Hz to 25 kHz. (E) and (F) Tonal response field maps from two additional locations in the macaque auditory cortex.

Also, did you get a different best frequency for two electrode contacts when the probe was in the same position ? Showing this (or better highlighting in the current figure) could help understand how useful the high resolution (small electrode pitch) is in differentiating best frequencies between close-by contacts.

Response: Given the tonotopic organization of the auditory cortex, and the orthogonal approach angle of the electrodes, the preferred frequency electrodes on the same shaft are likely to be very similar. The benefit of the close spacing for mapping of preferred frequencies could be highlighted by inserting the probes parallel to the superior temporal plane from posterior to anterior. In this case, frequency would be expected to change along the shaft.

However, benefits of the close spacing of the electrodes go well beyond the mapping of preferred frequency: (i) It will allow us to collect a larger number of single cells per insertion, thus increasing the cost-benefit ratio. (ii) Even if close-by single cells have similar preferred frequencies, they differ with respect to many other features, such as response latency. (iii) Pairwise connections between cells tends to be strongest for neurons that are close to each other; hence the closer spacing will facilitate the study of functional connections at the single-cell level; (iv) During acute recordings, the brain tends to settle over the course of the experiment, thus causing relative movement between the cells and the electrode contacts. If electrodes are spaced far apart, it is possible to lose neurons in between electrodes. Closer spacing allows a neuron to be simultaneously visible on more than one electrode, thus allowing them to be tracked as the brain settles.

Revision: The following text is added to the manuscript:

(Page 30): “Given the tonotopic organization of auditory cortex and the orthogonal insertion angle, adjacent contacts on the same shaft during each recording session showed very similar best frequencies; larger frequency gradients would emerge if steeltrode was inserted parallel to the superior temporal plane from posterior to anterior. Although we observed uniform best-frequency tuning across contacts at a single implantation site, future studies can leverage the steeltrode’s high-density array to resolve millisecond-scale response-latency differences among neurons that may share similar best frequencies. Additionally, this fine spacing will facilitate probing of local synaptic interactions, which are strongest among closely spaced cells. The successful acquisition of TRF maps using steeltrode demonstrates its effectiveness and strong potential for probing functional organization in deep brain structures, offering valuable insights for neuroscientific research.”

Figure 5 legend: Labels in legend C,D,E,F don't match figure.

Response: Thanks for pointing out the error. It is now corrected

Figure 5 E-F: Can the depth at which these were recorded be given ? Or at least the distance in the tissue between the two measurements ?

Response: The two recordings were made on separate days from different locations within auditory cortex, with distinct preferred frequencies, consistent with the frequency gradient observed in primary auditory cortex. The distance between the sites for Figures 6E and 6F relative to the site in Figure 6D is approximately 4.5 mm posterior for the site in 6E, and 6.0 mm posterior with an additional 1.5 mm lateral offset for the site in Figure 6F. The depth of the recordings was estimated based on the polarity inversion of the P1 component, which is believed to originate in deep layer 3. Given that the majority of cells are found in deep layer 3 and below, both recording sites are putatively located in deep layer 3.

Revision: The following text is added to manuscript to indicate the estimated recording location and depth:

(Page 30): TRF maps from two additional recording sessions—where the steelrode was implanted at different locations in auditory cortex—are shown in Figure 6E and 6F. These sites were located approximately 4.5 mm posterior, and 6.0 mm posterior with an additional 1.5 mm lateral offset, respectively, relative to the location corresponding to the TRF map in Figure 6D. These spatial offsets were calculated based on the known 1.5 mm pitch of the cortical positioning grid. To enable direct comparison across sessions, each electrode's MUA response was normalized by the maximum response observed across all frequencies, intensities, and electrodes within that session. This yielded a normalized scale from 0 to 1, allowing standardized comparisons across recordings. The resulting TRF maps reveal distinct best frequencies of 17.4 kHz and 20.5 kHz at the two more posterior sites (Figure 6E and 6F, respectively), reflecting regional variability in frequency tuning along the auditory cortex. Recordings were made at depths estimated to correspond to cortical Layer 3, based on microdrive depth tracking, known cortical thickness, and characteristic tone-evoked MUA profiles.

Figure 5: “normalized MUA response”: be more specific in the methods how this was computed and especially normalized.

Response: Thank you for your comment. For each stimulus condition (at a specific frequency and a specific volume intensity), the MUA response was calculated as the average power of the multi-unit activity signal within a defined temporal window (15–50 ms post-stimulus onset) for a given channel. To facilitate comparison across conditions, the MUA response for each channel was normalized by dividing by the maximum MUA response observed over **all** stimulus frequencies and intensities for all the active recording channels in that recording session. This normalization, yielding a relative scale from 0 to 1, allows us to directly compare TRF maps obtained from different locations in the auditory cortex, across separate recording sessions with steelrodes implanted at different locations.

Revision: The following text is added to the manuscript:

(Page: 30): “To enable direct comparison across sessions, each electrode’s MUA response was normalized by the maximum response observed across all frequencies, intensities, and electrodes within that session. This yielded a normalized scale from 0 to 1, allowing standardized comparisons across recordings.

Figure 5D: Y-label “Response” ? Is the also the normalized MUA response?

Response: The y-label has been updated for clarity. It is also a normalized MUA response.

Discussion, P.33: “as well as long-term chronic brain interfacing”: there is no demonstration of chronic use of these probes here, unless I missed it. I would discuss this as a future next step, but not as something readily demonstrated here.

Response: We have revised the text accordingly to address this comment.

Revision: Here is the updated text:

(Page 37): “This design is particularly promising for chronic implants, as the flexible tether cables accommodate brain micromotions and can significantly reduce tethering forces, potentially minimizing chronic tissue damage. These enhancements in hybrid architecture are expected to be valuable for future studies aimed at optimizing probe performance and demonstrating steeltrode’s long-term recording performance.”

Discussion, P.33: “Using selective electrochemical etching of stainless steel, we have created completely flexible tether cables attached to rigid shuttles and demonstrated hybrid rigid-flexible probe architectures for non-human primates. Such devices can be implanted into deep cortical regions, while being free to move with brain micromotions ensuring minimized chronic tissue damage due to significantly reduced tethering force.” This was fabricated, but not implanted and tested (or at least not shown in this manuscript). I would more clearly mention that this would be an interesting improvement to the architecture used in the present NHP recordings, especially for chronic implants, and that is technically feasible, and could/should readily be tested.

Response: Thank you for your feedback. We have revised the *Discussion* section (see the previous revision on Page 37) to suggest that these hybrid architectures show promise for chronic applications, and warrant investigation in future studies.

Discussion, P.33: “different depths of auditory cortex in macaque monkeys”. If n = 1 monkey, keep “monkey” singular.

Response: The text has been corrected.

Figure S2: “metal”: can you be more specific as to which ones ?

Response: The metal stack (Pt/ Au/ Pt) is specified in the figure.

Figure S3: (c) legend should be labeled as (b).

Response: The figure is updated and the legend is corrected.

What about stimulation using the steeltrode ? Could this be achieved ? Has it been tested ? It could be interesting to mention/discuss this in the discussion. To the authors' discretion.

Response: Stimulation using the steeltrode is very feasible, although we have not yet tested this application. Our electrodes are made of platinum, and we have already demonstrated the use of PEDOT:PSS coatings, which are known to enhance charge injection capacity and reduce impedance—key factors for effective electrical stimulation. Additionally, the platinum surface is amenable to other surface modification techniques that can further improve and stabilize stimulation performance. This promising potential for stimulation is definitely interesting and highly relevant for future investigations.

Revision:

The following text is added to the *Discussion* section of the manuscript:

“While we have not yet tested electrical stimulation with steeltrodes in this study, their platinum electrodes—especially when coated with PEDOT:PSS—offer high charge injection capacity and low impedance, supporting effective stimulation. Moreover, our fabrication process is compatible with SIROF and other surface modification strategies, paving the way for robust, integrated electrical stimulation capabilities in future work.”

References:

Main, P.2: missing some references to support the claim that very small cross-section is important to minimize tissue damage.

Response: A reference (Otte, E., Vlachos, A. & Asplund, M. Engineering strategies towards overcoming bleeding and glial scar formation around neural probes. *Cell Tissue Res* 387, 461–477 (2022)) is added to the corresponding text that argues that the cross-sectional footprint is a critical factor in reducing gliosis and inflammation.

Main, P.2: Ref. 13 and 14 are actually the same. Also, more recent literature could be added and interesting to refer to: <https://doi.org/10.1016/j.jneumeth.2023.110016>

Response: Thanks for pointing it out. A couple of recent articles (including the one recommended by the reviewer) is cited.

Figure 4: Could be interesting to put this into perspective/discuss with more recent literature using Plexon stainless steel electrode: <https://doi.org/10.1016/j.neuroimage.2023.120364>

Response: We have added reference to this paper in our manuscript (discussion section) which uses a 150 micron pitch Plexon probe to develop a model to predict neural circuit response in auditory. The much higher density that steeltrode provides (with 25 micron pitch electrodes), along with the customizability of the electrode arrangements facilitate development of even more accurate modeling of such neural dynamics through detecting closely-spaced neurons that may be missed using the conventional neural probes. Furthermore, the high degree of customizability in electrode arrangement in steeltrode—enabled by precise photolithographic and microfabrication processes—allows for probe designs tailored to the specific spatial and anatomical requirements of neuroscience experiments, such as the targeted interrogation of different brain regions. We have added a brief discussion to the manuscript to highlight this point and put our work in the context of the state of the art.

Revision: The following text is added to the *Discussion* section.

“Compared to manually assembled commercially available stainless steel probes for non-human primate experiments, which are limited to a maximum of 64 channels with a minimum pitch of 50 μm , our microfabrication approach enables significantly higher channel densities with extreme customizability. We have demonstrated electrode pitches as low as 25 μm , allowing for a much higher spatial resolution. Furthermore, our fabrication process supports feature sizes as small as 1 μm , which facilitates electrode counts of 100 channels per single metal layer. This number can be further scaled up using the multilayer microfabrication approach presented in this work, allowing for an even greater number of recording sites without significantly increasing the probe footprint. We strongly believe that ultra-high spatial resolution recording from deep brain regions facilitated by steeltrode (with electrode pitch down to 25 microns) holds significant potential for improving cortical circuit models compared to those developed with commercial stainless steel probes featuring a 150 μm pitch, as reported in by Chien et. al. Moreover, the high degree of customizability in electrode arrangement on the steeltrode platform—enabled by precise photolithographic and microfabrication processes—allows for the design of probes tailored to neuroscience experiments that require targeted interrogation of specific and isolated brain regions.”

My expertise:

Sections regarding the microfabrication and micromachining of the steeltrode themselves and how novel the approach is, are out of the scope of my expertise. I leave it to the other reviewers to judge the feasibility, soundness and novelty of this process.

We appreciate the constructive feedback provided by the reviewers and the editorial team. We have revised the manuscript and carefully addressed all the concerns and suggestions raised. Revisions made to the manuscript are highlighted with blue colored text.

Reviewer 1:

I appreciate and thanks to authors about the responses to my comments. All of my concerns have been adequately addressed, and I have no further suggestions. I support the publication of the revised manuscript.

Response: We thank the reviewer for the thoughtful feedback and kind words. We are pleased to know that all concerns have been adequately addressed and that the revised manuscript meets your expectations.

Reviewer 2:

Thank you for addressing my concerns. I believe the manuscript has gained in clarity and additional data that was previously missing to support some of the main claims.

I have only a few additional small comments/remarks:

P.6 last paragraph end: “while minimizing the tissue damage. Damage.” : Typo, twice “damage”.

P.7 I.3: “the buckling force for the will be 40mN”: word missing after “the”.

P.9 I.21: “For steeltrodes, the devised microfabrication ...”: “devised”?

Response: Thank you for pointing these out. These errors are now corrected in the revised manuscript.

P.22: “Error reference not found” x2

Response: Thank you. We have corrected the figure citation error.

Legend Fig. 8 C) / P.34 I. new paragraph / “Statistical Power, Replication, and Exclusion Criteria” section: Can you make more clear in all these sections what N you are mentioning (and make sure you mention the appropriate N): is it N = number of animals or N = number of insertions or N = number of cross sections, and how many insertions/animals/sections per group/condition (control, dura removal, dura piercing) for comparison analysis? You mention in your response “The histology data were collected from 5 different insertions in 3 animals. Two insertions were performed in each animal.”: was the second insertion performed on another hemisphere for example to avoid tissue effect due to the first insertion when looking at histology ? What happened to the 6th insertion (if 2 insertions/animals) ?

Response : We thank the reviewer for this important comment and apologize for the earlier ambiguity regarding sample sizes and experimental design. We have now revised the manuscript to provide a clear and detailed account of the number of animals, hemispheres, insertions, and histological sections used in each condition. Specifically, we clarify that all insertions were performed in 5 hemispheres and no insertion was performed in 3 hemispheres from a total of 4 animals. The hemispheres were allocated as follows: 3 for control (no insertion), 2 for dura removal, and 3 for dura piercing. From each hemisphere, 6-9 coronal sections were analyzed, and statistical comparisons were conducted at the section level. We have also corrected and added justification for the use of a total 4 animals (including control) in this exploratory study. These revisions are reflected in the updated *Methods* section under “Statistical Power, Replication, and Exclusion Criteria (Rodent).”

Reference 30 and 56 are the same ?

Response: Thank you for pointing this out. This error is now corrected in the manuscript.

Fig. 8 A: can a scale bar be added for the images ?

Response: The scale bars have been added to the images.

Reviewer 3:

The authors have provided a detailed and systematic procedure on fabrication of steeltrode used as a brain sensor. The design aspects of the probe seem fine in my view, though following minor issues may be considered.

- Though fabrication steps have been provided in details, still it is not very clear that how did the authors made the long shank which is inserted in the brain. some clear figure may be provided.

Response: Thank you for your comment. We have included an additional figure (Figure S3) to the supplementary material to clarify the steeltrode release process.

To fabricate steeltrode in a high throughput process, we use planar microfabrication processes on stainless steel wafers to deposit insulation layers, deposit and pattern high-density metal traces, and define selective openings to expose the recording electrodes, backend contact pads, and probe outlines. The details of these planar microfabrication processes is discussed in the *Methods* section.

After wafer-level processing is completed, we employ a laser micromachining process to release the steeltrodes with long implantable shanks from the stainless-steel wafer. In this step, the outline of each steeltrode device is precisely etched using a 200 W fiber laser through laser ablation. The laser is precisely guided along the device (steelrode) boundaries based on a CAD layout, using alignment fiducials patterned on the wafer during the planar microfabrication process. An example of the laser etching path for a steeltrode on a processed stainless steel wafer is shown in Figure S3. By repeating this automated process for each steeltrodes on a wafer, the individual

steelrodes are cleanly released from the substrate. Representative images of the released steelrodes following laser micromachining are also shown in Figure S3.

- Also, in the figure 2S the first and second figure are same. Please look into this.

Response: Thank you for pointing this out. This figure is now corrected in the manuscript.